## OPEN

# Sugars dominate the seagrass rhizosphere

E. Maggie Sogin [1,2 ✉], Dolma Michellod[1], Harald R. Gruber-Vodicka [1], Patric Bourceau[1,3], Benedikt Geier[1], Dimitri V. Meier [4], Michael Seidel [5], Soeren Ahmerkamp [1], Sina Schorn [1], Grace D'Angelo [1], Gabriele Procaccini [6], Nicole Dubilier [1,3 ✉] and Manuel Liebeke [1 ✉]

**Seagrasses are among the most efficient sinks of carbon dioxide on Earth. While carbon sequestration in terrestrial plants is linked to the microorganisms living in their soils, the interactions of seagrasses with their rhizospheres are poorly understood. Here, we show that the seagrass, *Posidonia oceanica* excretes sugars, mainly sucrose, into its rhizosphere. These sugars accumulate to μM concentrations—nearly 80 times higher than previously observed in marine environments. This finding is unexpected as sugars are readily consumed by microorganisms. Our experiments indicated that under low oxygen conditions, phenolic compounds from *P. oceanica* inhibited microbial consumption of sucrose. Analyses of the rhizosphere community revealed that many microbes had the genes for degrading sucrose but these were only expressed by a few taxa that also expressed genes for degrading phenolics. Given that we observed high sucrose concentrations underneath three other species of marine plants, we predict that the presence of plant-produced phenolics under low oxygen conditions allows the accumulation of labile molecules across aquatic rhizospheres.**

Seagrasses are marine angiosperms that form the foundation of ecologically and economically valuable ecosystems along coastal regions of all continents except Antarctica[1]. As ecosystem engineers, seagrasses provide important services to humans. For example, seagrass meadows are important habitats for fisheries, stabilize the sea floor, remove pollutants and drive biogeochemical cycling[2]. Seagrasses are highly efficient in sequestering carbon from the atmosphere by incorporating it into their tissues and burying it as organic matter in their sediments[3]; they bury carbon 35 times faster than rainforests per unit area[4]. Seagrasses also excrete high concentrations of dissolved organic carbon (DOC) into the environment[5], which is assumed to be partially metabolized by the microorganisms in the sediments surrounding seagrass rhizomes and roots.

On land, angiosperms excrete up to 30% of their primary production as organic compounds to their soils to attract beneficial microbial partners, to defend themselves from pathogens and to communicate with other individuals[6]. These root exudates feed complex microbial food webs that drive the long-term storage of organic carbon[7]. Much less is known about how seagrasses interact with the microbial communities in their sediments[8]. Here, we show that seagrass sediments are veritable sweet spots in the sea that contain surprisingly high concentrations of simple sugars, primarily in the form of sucrose. Our results indicate that microbial consumption of these reactive sources of organic carbon is inhibited by the presence of phenolic compounds in the seagrass rhizosphere.

## Results and discussion

***Posidonia oceanica* sediments are rich in sugars.** Our metabolomic analyses of pore water chemistry underneath a meadow of the endemic Mediterranean seagrass, *P. oceanica*, unexpectedly revealed concentrations of sugars in the high μM range (Fig. 1a,b, Supplementary Fig. 1 and Supplementary Tables 1 and 2). The

primary sugar in the pore water metabolome was the disaccharide sucrose (Supplementary Fig. 2), which also has been shown to be the most abundant sugar in the tissues of seagrasses, including *P. oceanica*[9]. Given that seagrasses occupy 300,000–600,000 km² of coastal regions[3], we conservatively estimated a global stock of 0.67–1.34 Tg of sucrose in the upper 30 cm of seagrass sediments. The sucrose concentrations we measured underneath *P. oceanica* are as much as 80 times higher than previously measured dissolved total carbohydrate pools from marine sediments and 2,000-fold higher than free sugar concentrations in the open ocean[10,11]. Our observations represent a paradox in microbial ecology, as it is widely known that most microorganisms quickly consume sugars, like sucrose, in their immediate environment[12,13]. Until now, the only known microbial habitats that contain similarly high concentrations of simple sugars include saps and fruit nectars, rhizospheres of agro-ecosystems including sugar beets and biofilms on mosses and marine macroalgae[14,15].

To understand sugar distribution across spatial scales, we collected 570 metabolomic profiles from sediment pore water underneath, 1 m and 20 m away from a *P. oceanica* meadow in the Mediterranean (in, edge and out; Fig. 1a). Non-targeted and targeted analyses revealed that sediments underneath *P. oceanica* contained significantly higher concentrations of sucrose, glucose, trehalose, myo-inositol and mannitol compared to the sites away from the seagrass (Extended Data Fig. 1, Supplementary Fig. 1 and Supplementary Tables 2 and 3). Bulk measurements of DOC were at least 2.6× higher inside the meadow than at the edge or outside (Extended Data Fig. 2a and Supplementary Tables 2–4). Measured sugars contributed up to 40% of the DOC pool underneath seagrass meadows, which form an energy-rich source of carbon that has not yet been described or characterized (Extended Data Fig. 2b)[5]. Previous studies showed that seagrasses excrete organic carbon produced from CO₂ fixation into their sediments[16] and hypothesized that sucrose is one of the major components of the excreted carbon

[1]Max Planck Institute for Marine Microbiology, Bremen, Germany. [2]University of California at Merced, Merced, CA, USA. [3]MARUM—Center for Marine Environmental Sciences of the University of Bremen, Bremen, Germany. [4]Institute of Biogeochemistry and Pollutant Dynamics, Department of Environmental Systems Science, Swiss Federal Institute of Technology, Zurich, Switzerland. [5]Institute for Chemistry and Biology of the Marine Environment, University of Oldenburg, Oldenburg, Germany. [6]Stazione Zoologica Anton Dohrn, Napoli, Italy. ✉e-mail: esogin@ucmerced.edu; ndubilie@mpi-bremen.de; mliebeke@mpi-bremen.de

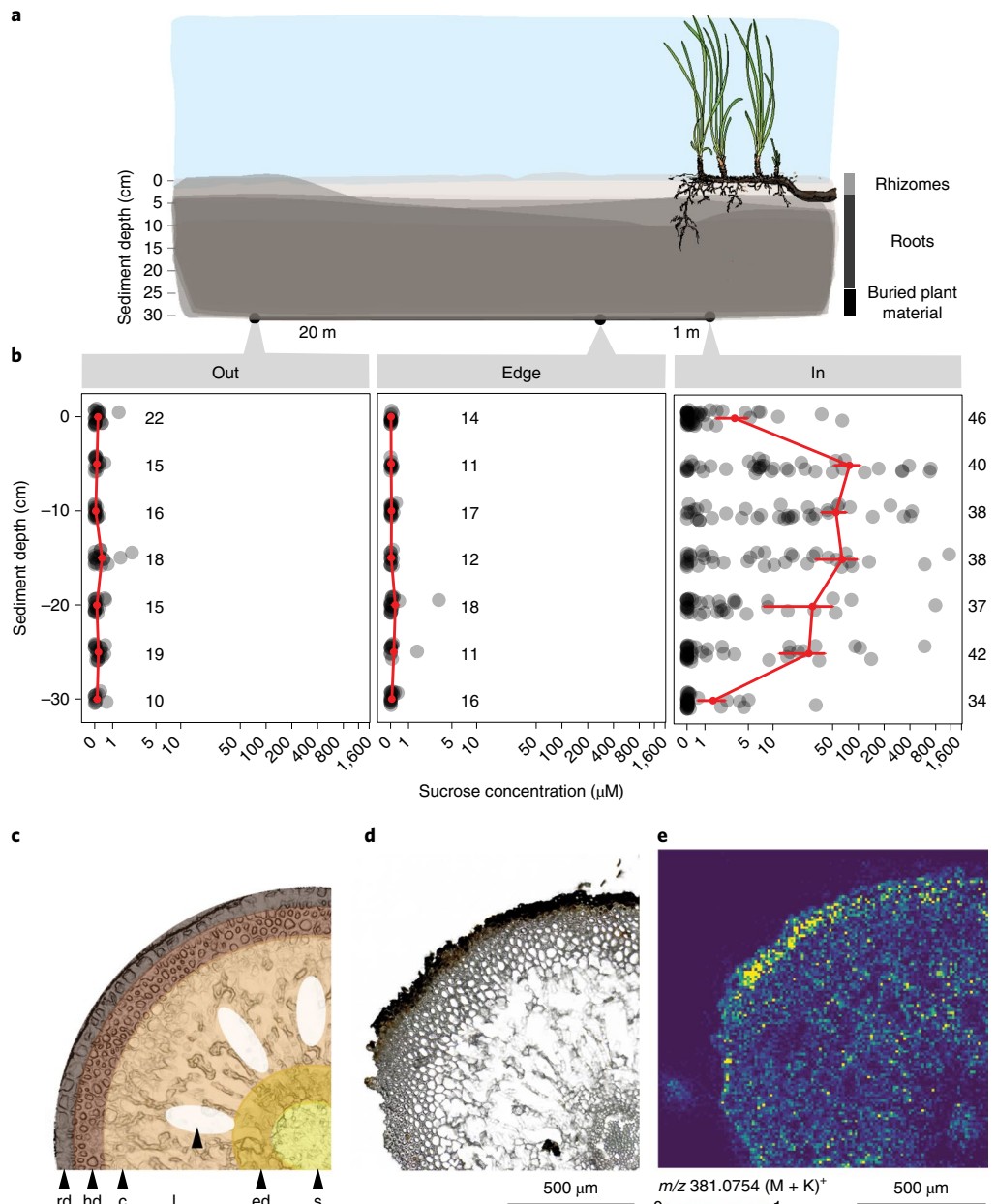

**Fig. 1 | Sugars were more abundant underneath seagrass meadows than in non-vegetated sediments. a**, Sediment pore water profiles were collected 20 m away, 1 m away and underneath a Mediterranean *P. oceanica* meadow off Elba, Italy. Pore water profiles consisted of samples collected every 5 cm from 0 to 30 cm sediment depth. **b**, Plots show sucrose concentrations in pore waters across sediment depth (replication numbers are reported for each location and sample depth combination). Red points represent mean concentrations ± s.e.m. Sample sizes at each sediment depth are indicated in the figure. Sucrose concentrations are plotted on a log scale. Sugar concentrations were significantly higher inside the meadow than outside or at the edge ($P = 3.56 \times 10^{-13}$; two-way ANOVA comparing sugar concentrations as a function of sediment depth and location; model results and significance values are reported in Supplementary Tables 2 and 3, average values across sediment depths are reported in Supplementary Table 1). **c–e**, Schematic, light microscopy (**c,d**) and matrix-assisted laser desorption/ionization MS images (**e**) from a *P. oceanica* root collected from a meadow in the Mediterranean off Elba, Italy. Displayed is a quarter of a root cross-section. In **e**, the molecular ion distribution of disaccharides ($C_{12}H_{22}O_{11}$, potassium adduct $[M + K]^+$, *m/z* 381.0754), including sucrose (as confirmed by GC–MS analyses), shows that the relative abundance was highest in the root rhizodermis. Replicate measurements ($n = 7$) of individual root sections are presented in Supplementary Fig. 5. Relative abundances are visualized as a heatmap from low (blue) to high (yellow) intensities of each pixel (10 μm). rd, rhizodermis; hd, hypodermis; c, cortex; l, lacunae; ed, endodermis; s, stele.

given its high concentrations in seagrass tissues[9]. However, until now the molecular composition of the organic carbon excreted by seagrasses was not known. Here, we provide evidence that sucrose is the major component of *P. oceanica* exudates and, in the following, link seagrass primary production to the accumulation of sucrose in the marine rhizosphere.

Through a series of temporal and spatial analyses, we provide evidence to support our assumption that sucrose enters the rhizosphere through *P. oceanica*'s roots. First, field experiments showed that sucrose concentrations differed across seasonal and daily time scales (Supplementary Text 1, Supplementary Fig. 3 and Extended Data Fig. 3) and were highest at the sediment depths where the

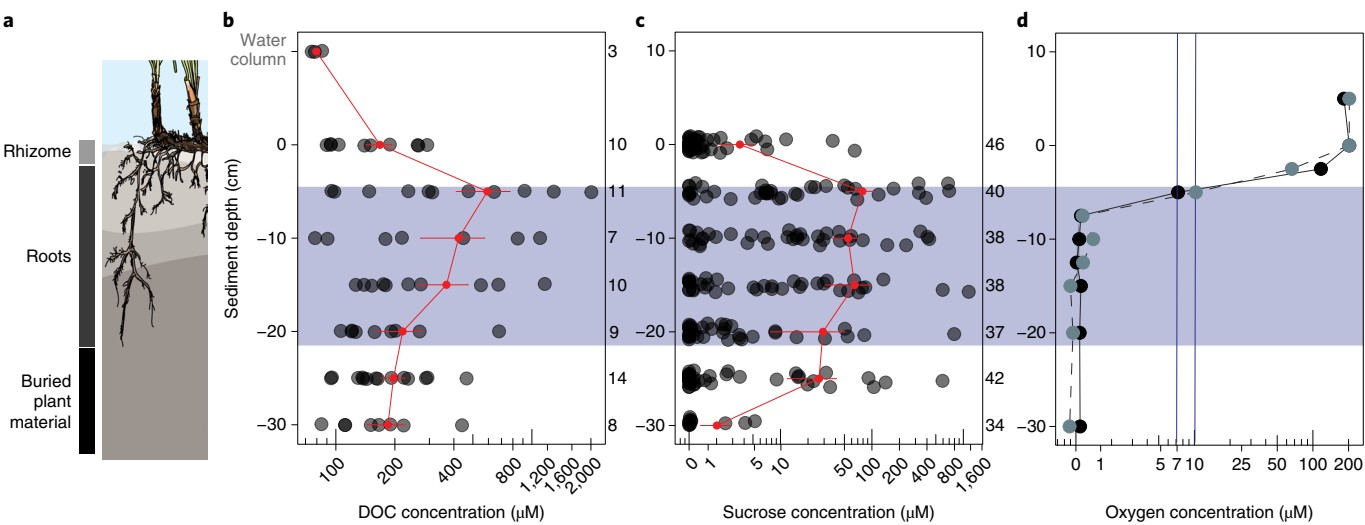

**Fig. 2 | Seagrasses deposit sucrose into their rhizosphere through their roots. a**, Schematic showing that *P. oceanica* roots extend between 5 and 20 cm into the sediments in Mediterranean seagrass meadows off the Island of Elba, Italy. **b,c**, DOC (Supplementary Table 4) (**b**) and sucrose concentrations (**c**) were highest in the zone of root penetration between 5 and 30 cm (two-way ANOVA $P < 0.001$; ANOVA model results and post-hoc tests reported in Supplementary Tables 2 and 3). Red points represent mean ± s.e.m. DOC and sucrose concentrations across sampling depth; red lines connect the means. Replication numbers across sediment depths are indicated next to **b** and **c. d**, Oxygen profiles ($n = 2$) using a lancet to measure $O_2$ in pore waters revealed that oxygen concentrations were highest at the sediment surface, decreased sharply between the upper sediment layers and 2.5 cm and became anoxic at 7.5 cm. Points represent sediment horizon depths where samples were collected; different line types and colours represent individual profiles.

seagrass roots occurred (Fig. 2a–c and Supplementary Fig. 4). Second, mass spectrometry (MS) images of cross-sections of *P. oceanica* roots revealed that sucrose was distributed throughout the root tissues. Sucrose was particularly abundant in the rhizodermis, the outermost cell layer of the root (Fig. 1c–e and Supplementary Fig. 5), suggesting that the sugar is excreted or passively released to the sediment through the rhizodermis. Both active transport and diffusion are known from terrestrial plants which release exudates, including sugars, through their roots into soils[6,17]. Collectively, our results from *P. oceanica* indicate that this marine plant transfers substantial amounts of sucrose to the surrounding sediments through its roots on a daily basis.

**Sucrose respiration is limited in meadow sediments.** Sucrose and other sugars are important carbon and energy sources for soil and sediment microorganisms. To understand why seagrass-produced sugars are not quickly consumed by sediment microorganisms, we measured in situ oxygen concentrations in sediments underneath a Mediterranean *P. oceanica* meadow. We observed a steep decrease in oxygen concentrations, from 202 μM to <0.5 μM, within the first 7.5 cm below the sediment surface reaching hypoxic conditions after 5 cm (7–10 μM, Fig. 2d). In our measurements, oxygen penetrated deeper than the mm depths commonly observed in sediments underneath and adjacent to *P. oceanica* meadows, although oxygen penetration depths down to 4 cm have been observed[18–20]. Past studies showed that seagrasses, including *P. oceanica*, release oxygen from their roots to their sediments[21] but we did not detect oxygen in deeper sediment layers, presumably because our oxygen sampling protocol could not detect variations in oxygen concentrations at scales <2.5 cm (Supplementary Methods).

The observed decrease in oxygen with depth underneath *P. oceanica* meadows suggests that microorganisms living in sediment layers deeper than 5 cm are not able to use aerobic pathways for sucrose respiration. In the absence of oxygen, microorganisms ferment sugars to organic acids, which a syntrophic community of bacteria and archaea respire to $CO_2$, mainly using nitrate and sulfate as electron acceptors. While nitrate is limited in sediments underneath seagrass

meadows[22], sulfate concentrations in Mediterranean pore waters exceed 28 mM and are stable across sites and sediment depths[20]. It is therefore likely that there was sufficient sulfate underneath *P. oceanica* meadows to allow sulfate-reducing microorganisms to metabolize the waste products of sugar fermentation. Coupled with previous observations of little to no sulfide underneath these meadows despite the availability of sulfate[20], we hypothesize that other biogeochemical factors beyond electron acceptors inhibit the microbial degradation of simple sugars underneath seagrass meadows.

Phenolics are one class of compounds that could limit the microbial degradation of sucrose under anoxic conditions, as shown for ruminal microbiomes where the presence of phenolics inhibits microbial growth[23]. To test our hypothesis, we analysed the molecular composition of sediment pore water dissolved organic matter (DOM) underneath and adjacent to a Mediterranean *P. oceanica* meadow using ultrahigh-resolution MS[24]. We found that, across sampling locations, 10–16% of all detected molecular formulae consisted of polyphenols and other highly aromatic compounds (molecular identification according to ref. [25]) (Fig. 3a). The composition of these DOM profiles was comparable to ecosystems with strong inputs of terrestrial organic matter, such as rivers[26,27]. *P. oceanica* contains phenolic compounds in its root tissues, including chicoric acid and caffeic acid (Extended Data Fig. 4 and Supplementary Fig. 6)[28]. Indeed, the sum formula for caffeic acid ($C_9H_8O_4$) had highest counts in the pore waters underneath the seagrass meadow between 5 and 25 cm sediment depth (Fig. 2b and Supplementary Table 2; see Supplementary Fig. 7 for identification of caffeic acid in sediment pore waters). Likely sources for caffeic acid in the pore waters include the active release from living seagrass or leachates from decaying *P. oceanica* material. Given that *P. oceanica* peat persists in sediments for hundreds to thousands of years and that decaying plant material is a source of lignin and phenolic compounds to soils on land, it is likely that decaying seagrass material is one of the major sources of phenolics in sediments underneath and next to the seagrass meadow[29,30].

To test if the presence of seagrass phenolics restricts microbial degradation of sucrose, we extracted phenolics from *P. oceanica*

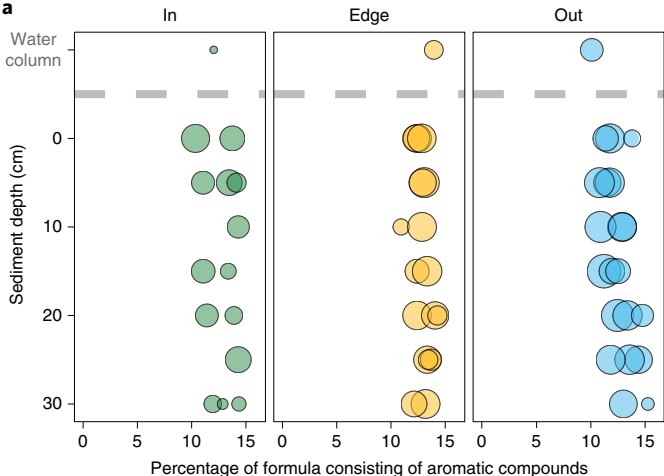

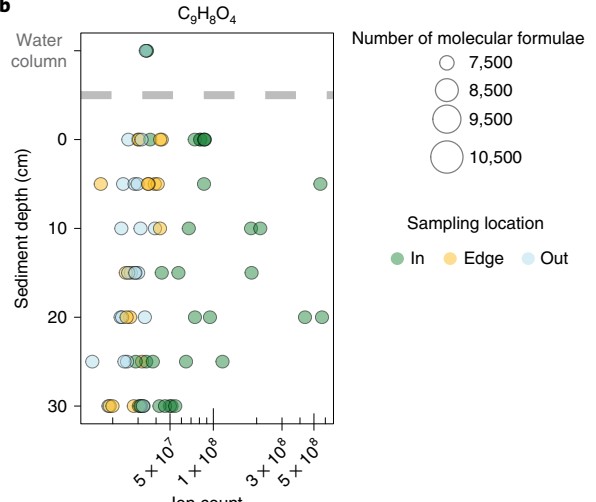

**Fig. 3 | Phenolics were present in sediment pore waters underneath and adjacent to a *P. oceanica* meadows. a**, Pore water profiles of the molecular DOM composition underneath (in) and 1 m (edge) and 20 m (out) from seagrass meadows, measured by ultrahigh-resolution MS. Our analyses revealed abundant aromatic molecular formulae, including polyphenols, which comprised 10–16% of all molecular formulae across sampling sites. Size of points reflects number of molecular formulae per sample. **b**, Ion counts of the molecular formula $C_9H_8O_4$, which includes the phenolic compound most abundant in seagrass tissues caffeic acid, are significantly higher inside the meadow than at the edge or outside the meadow (two-way ANOVA, location $P = 0.00026$). Count distributions also reveal higher counts within the root–sediment interface, between 5 and 25 cm below the sediment surface (two-way ANOVA, location × depth $P = 0.03$). Identification of caffeic acid ($C_9H_8O_4$) in pore waters beneath *P. oceanica* was confirmed using LC–MS/MS (Supplementary Fig. 7). For statistical results and post-hoc tests see Supplementary Tables 2 and 3.

tissue (Supplementary Fig. 6) and co-introduced this extract with $^{13}C_{12}$-sucrose to meadow sediments under oxic and anoxic conditions using slurry experiments (Supplementary Text 2). We observed that the presence of seagrass-derived phenolics under anoxic conditions significantly decreased $^{13}C$-sucrose respiration rates and consequently the production of $^{13}CO_2$ over tenfold. In contrast, under oxic conditions we saw no significant effect of phenolics on sucrose respiration (Fig. 4a,b and Supplementary Text 2). We used the measured respiration rates from our anoxic sediment incubations to compare how quickly the calculated standing stock

of sucrose underneath a *P. oceanica* meadow of 3.99 mmol m$^{-2}$ in the anoxic sediment layers (5–20 cm) could be completely respired to $CO_2$ in the absence and presence of seagrass-derived phenolics (for calculations see Supplementary Table 5 and Supplementary Text 3). In the absence of phenolic compounds under anoxia, the standing sucrose pool underneath the meadow would be completely respired to $CO_2$ at a rate of 232 mmol(C) m$^{-2}$ d$^{-1}$ in seawater (Fig. 4c), resulting in the complete removal of sucrose to $CO_2$ in less than one day (Supplementary Table 5). When we consider the presence of seagrass-derived phenolics, our incubation results showed that under anoxia, phenolics decreased respiration rates of C from $^{13}C$-sucrose to 20 mmol(C) m$^{-2}$ d$^{-1}$ (Fig. 4c and Supplementary Table 5). Our results are consistent with previous studies showing that phenolic inhibition of microbial respiration increases with decreasing oxygen concentrations[31–33]. They are also consistent with the requirement for oxygen by microbial phenol oxidases (PO), the enzymes central to the microbial oxidation of phenolic compounds. Studies exploring PO activity in peatlands showed that phenolic degradation decreases with decreasing oxygen availability[31,34]. In *P. oceanica* meadows, the degradation of phenolics in the oxic layer under the meadow would allow the sediment microbial community to fully respire sucrose, while under low to no oxygen conditions the high concentrations of phenolics inhibit microbial consumption of sucrose and sugars accumulate under the seagrass meadows.

Our incubation experiments show that the microbial community underneath seagrass meadows have the metabolic ability to degrade sucrose but this potential is significantly reduced by the presence of phenolics in their environment. Our work, together with research on plant exudation in terrestrial rhizospheres, indicates that both marine and land plants excrete sugars and phenolic compounds to their soils[30,35]. However, in terrestrial rhizospheres where there is sufficient oxygen, these phenolic compounds are degraded[36]. This removes the inhibitory effect of most phenolic compounds on microbial metabolism, so that microbial communities are able to consume plant-produced sugars[37]. In contrast, in the marine rhizosphere of *P. oceanica*, only the first few centimetres underneath the meadow are oxic, while most of the sediment underneath the meadows is anoxic. In these anoxic sediments, seagrass-produced phenolics are not quickly oxidized and thus accumulate, limiting the microbial degradation of sucrose and allowing sugars to accumulate in the seagrass rhizosphere. Thus, the presence of phenolics and the lack of oxygen underneath seagrasses delays the input of reactive, organic carbon into microbial respiration, allowing sucrose to accumulate.

**Most seagrass rhizosphere microorganisms do not use sucrose.** Plant carbon deposition to terrestrial soils serves as a chemo-attractant for beneficial microorganisms, including arbuscular mycorrhizas and beneficial bacteria and archaea[6]. We hypothesize that seagrass deposition of sucrose, phenolics and other compounds into sediments, structures the composition and function of the associated marine rhizosphere.

Using metagenomics and full-length 16S rRNA amplicon sequencing from sediments collected inside and 1 m and 20 m outside the meadow, we show that the sediments underneath seagrass meadows contained a distinct community of microorganisms (Fig. 5a,b and Extended Data Fig. 5). Using principal coordinate analyses (PCoA), samples clustered according to their collection site, on the basis of the taxonomic composition of both metagenomic and full-length 16S rRNA sequencing libraries (Fig. 5a,b). Our amplicon sequencing further revealed that bacterial communities were highly diverse across collection sites, mirroring known complexities in terrestrial habitats that contain thousands of taxa per gram of soil[38]. However, we observed fewer amplicon sequence variants (ASVs) inside than outside of the meadow (Fig. 5c). These

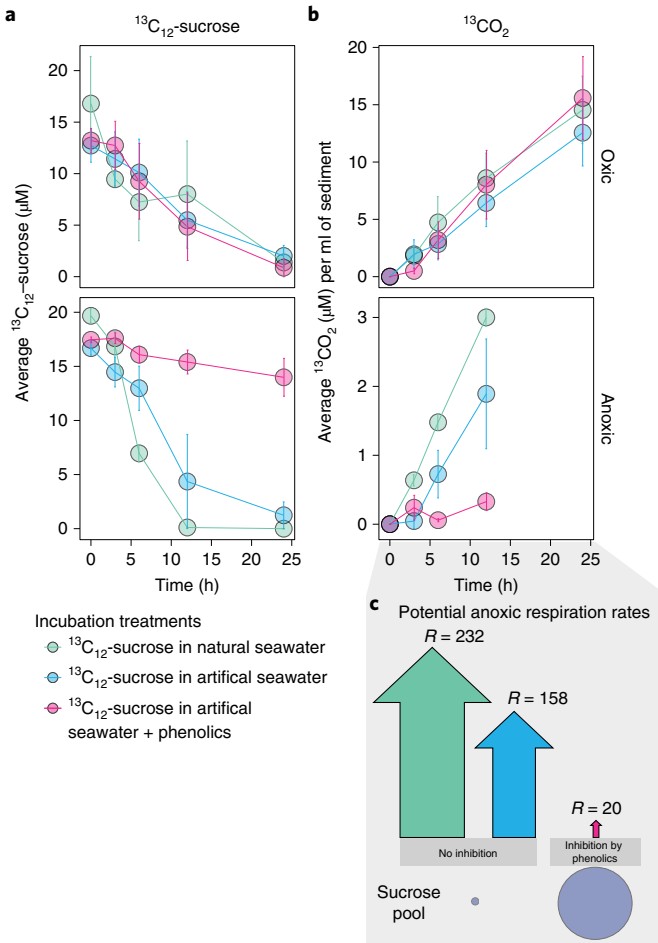

**Fig. 4 | Phenolic compounds inhibit the degradation of sucrose.**
**a,b**, Sediments from replicate cores ($n = 3$) collected from inside a Mediterranean *P. oceanica* meadow off the Island of Elba, Italy, consumed $^{13}C_{12}$-sucrose (**a**) and produced $^{13}CO_2$ (**b**) over the course of oxic and anoxic experiments conducted over 24 h. Individual points in both **a** and **b** represent treatment means ± s.e.m. across independent replicates ($n = 3$). Under oxic conditions, the production rate of $^{13}CO_2$ from $^{13}C_{12}$-sucrose was similar across treatments and at least 2.5× higher than rates observed under anoxia (note different *y* axis scales). However, in the anoxic incubations, the addition of phenolics extracted from *P. oceanica* tissues inhibited the microbial degradation of sucrose to $CO_2$. **c**, The estimated measured respiration rates under anoxia show that the production of $^{13}CO_2$ from sucrose in the presence of phenolic compounds is at least eight times lower than that of natural or artificial seawater condition (Supplementary Table 5). The size of the arrows is proportional to the potential rate of $CO_2$ released from sucrose in mmol(C) m$^{-2}$ d$^{-1}$.

data indicate that the presence of the seagrass rhizosphere not only selects for specific taxa but also limits the diversity of the community. Similarly, on land, plants secrete compounds to their rhizospheres, creating a habitat that shapes the community composition of their soil taxa[39] and results in a specialized microbial community with lower diversity than neighbouring unvegetated soils[40,41].

Microbial genomes recovered from our metagenomic analyses (Supplementary Fig. 8) revealed that taxa in the *P. oceanica* rhizosphere appear to be limited in their ability to degrade sucrose in spite of the high abundance of this simple sugar in their environment. We recovered 182 metagenomic assembled genomes (MAGs), 125 of which were of medium draft quality (Supplementary Tables 6 and 7) and focused our metagenomic and metatranscriptomic analyses

on genes used in microbial sucrose metabolism. Although 80% of the recovered MAGs contained genes that encode for sucrose degradation, including sucrose phosphorylases and sucrose-6-phosphate hydrolases, these genes were expressed in only 64% of the MAGs (Supplementary Table 8). Furthermore, for most MAGs, the expression of genes that encoded for sucrose degradation were not among the top 10% of all expressed orthologues (Supplementary Text 4 and Supplementary Fig. 9).

Despite the relatively low expression of genes that encode for sucrose degradation across most MAGs, we identified six MAGs across sampling sites that expressed a higher proportion of genes encoding for sucrose degradation compared to other sugars ($TPM_{sucrose}/TPM_{other} > 1$; Fig. 5d and Extended Data Fig. 6). On the basis of our genomic analyses, we predict that three of these six bacterial species preferentially metabolize sucrose over other sugars inside the seagrass meadow. These putative sucrose specialists were undescribed members of the Xanthomonadales (MAG 142), Verrucomicrobiales (MAG 209) and environmental gammaproteobacterial lineage UBA6522 (MAG 438) (Fig. 5e–g). The putative sucrose specialists at the edge of the meadow belonged to the Desulfobacterales (MAG 154), Thiohalomonadales (MAG 207) and Beggiatoales (MAG 76) (Fig. 5h–j). In addition to sucrose degradation, these six MAGs covered a range of metabolic pathways, including fixation of $CO_2$ and $N_2$, the use of reduced sulfur compounds and hydrogen as energy sources and sulfate respiration (Supplementary Table 8 and Supplementary Text 4). Both MAGs 76 and 154 are facultative anaerobes on the basis of the absence of cytochrome *c* oxidase in their genomes.

Given our experimental results showing that phenolic compounds inhibited microbial sucrose degradation under anaerobic conditions, we assume that anaerobic sucrose degraders should be able to degrade phenolics. Indeed, expression data indicated that both MAG 76 and MAG 154 degrade phenolics through the benzoyl-CoA and beta-ketoadipate pathways (Supplementary Table 8). Furthermore, expression data for three of the four aerobic MAGs indicated that these also degrade phenolics, allowing them to metabolize sucrose in the upper layers of the seagrass rhizosphere (Supplementary Text 4). It is intriguing that the $N_2$-fixing symbiont in the root tissues of *P. oceanica*, which also gains its energy and carbon from plant-derived sugars, expresses genes involved in detoxifying phenolic compounds[19], indicating the strong selective advantage this trait provides for seagrass-associated microorganisms.

Our metagenomic and metatranscriptomic results are consistent with our sediment incubation experiments. Many members of the microbial community underneath the meadow have the genomic ability to degrade sucrose. However, low transcription of genes involved in sucrose metabolism, together with our measurements of high sucrose concentrations underneath the meadow, indicate most members of the seagrass rhizosphere do not actively degrade this carbon source. We hypothesize that the few taxa that disproportionately expressed genes for sucrose degradation over other sugars evolved mechanisms that enabled them to thrive on this abundantly available and easily degradable energy and carbon resource in the seagrass rhizosphere despite the presence of phenolic compounds. This is consistent with the wealth of data from terrestrial rhizospheres showing that select microorganisms are able to use sugars and other compounds in the presence of phenolics[36,37,42]. In fact, in terrestrial rhizospheres, phenolic compounds work to shape soil community composition by acting as antimicrobials towards some microbial taxa while stimulating others[42,43].

**Sucrose persists in other aquatic rhizospheres.** We hypothesize that other types of aquatic rhizosphere also contain high concentrations of sucrose and other sugars, given that most plants produce phenolic compounds and oxygen is rapidly depleted in the first few

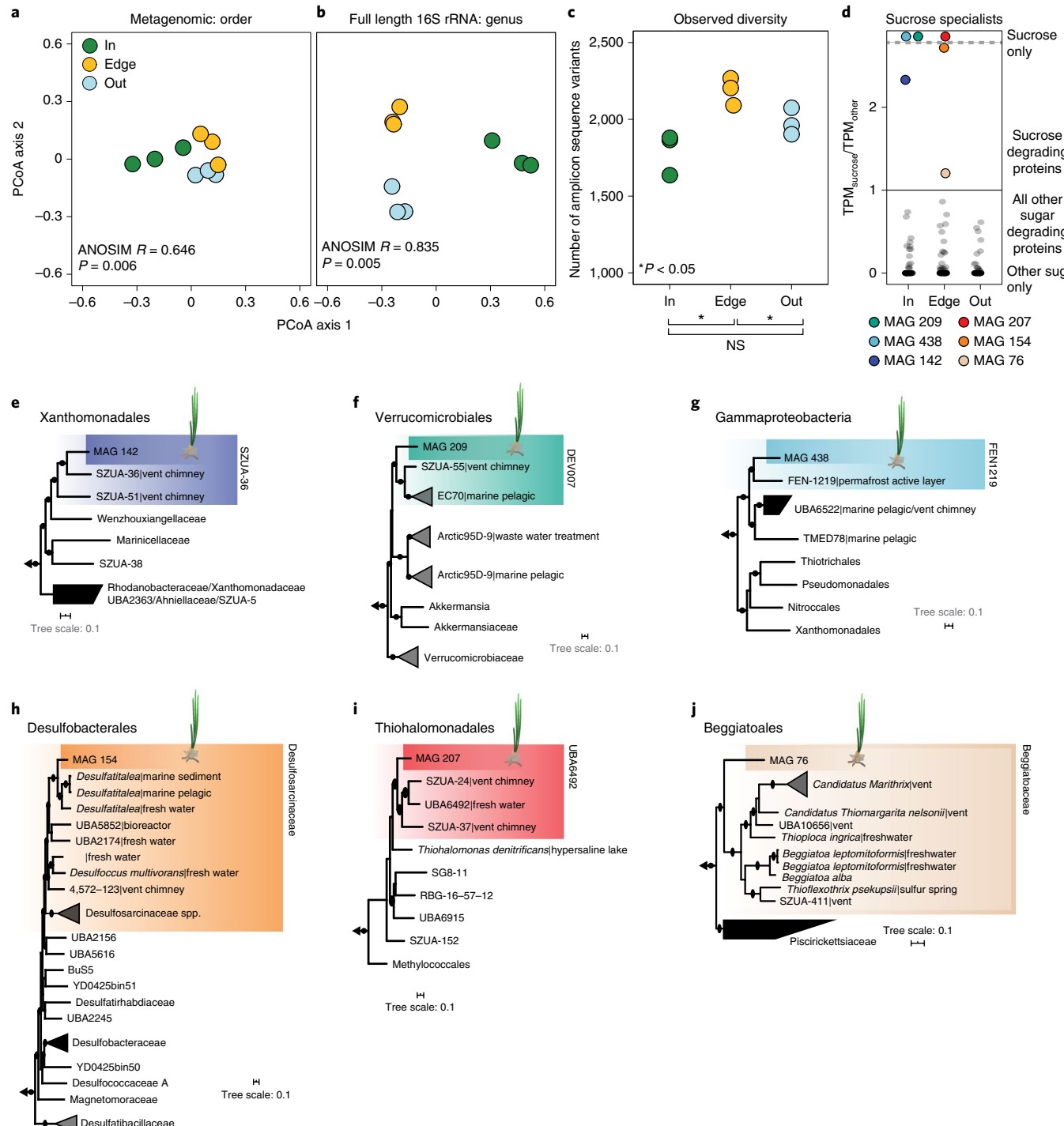

**Fig. 5 | Microbial community composition and metabolism were specific to seagrass sediments. a,b**, PCoA analyses show that the taxonomic composition of microbial communities underneath (in) and away (edge and out) from a Mediterranean *P. oceanica* meadow off Elba, Italy, were significantly different on the basis of bulk metagenomics (order level, one-way analysis of similarities (ANOSIM) $R^2 = 0.65$, $P = 0.006$) (**a**) and full-length 16S rRNA amplicon sequencing (genus level, one-way ANOSIM $R = 0.835$, $P = 0.005$) (**b**). **c**, The total number of ASVs was significantly lower in samples collected inside and outside the meadow than at the edge ($M_{in} = 1,793 \pm 79$ s.e.m.; $M_{edge} = 2,224 \pm 23$ s.e.m.; $M_{out} = 1,983 \pm 51$ s.e.m.). NS, not significant. **d**, Boxplots show the accumulative expression ratio in each MAG between glycoside hydrolases (GH) predicted to degrade sucrose compared to other sugars (overlying points). Putative sucrose specialists (coloured points) are defined as having higher TPMs for sucrose degradation compared to all other GH enzymes ($TPM_{sucrose}/TPM_{other} > 1$). TPMs represent the total expression across each site-specific library. The majority of MAGs across sites were sugar generalists, while putative sucrose specialists were only found in three MAGs inside and three at the edge of the meadow. **e–j**, Phylogenomic trees for six sucrose specialists: MAG 142 (**e**); MAG 209 (**f**); MAG 438 (**g**); MAG 154 (**h**); MAG 207 (**i**); and MAG 76 (**j**). Coloured boxes show members that grouped within the same family clade. For the relative abundance of each of the putative sucrose specialists across habitats see Supplementary Fig. 10.

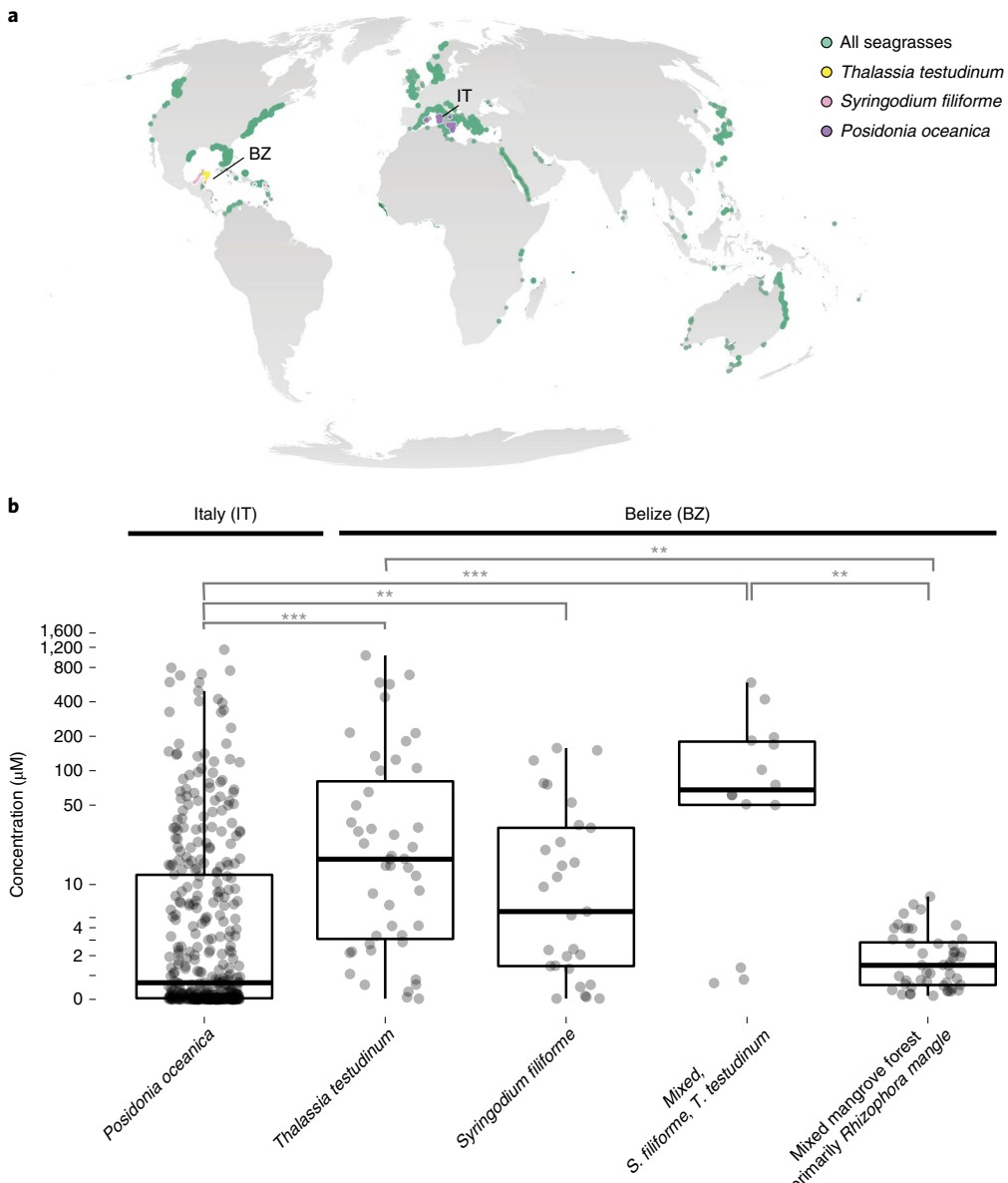

**Fig. 6 | Sediment pore waters underneath seagrass meadows and a mangrove forest contained high concentrations of sucrose. a,** Seagrasses occur in coastal waters worldwide (data accessed from UN Environmental World Conservation Monitoring Center on 21 March 2020, http://data.unep-wcmc.org/datasets/7). *S. filiforme* (pink), *T. testudinum* (yellow) in Belize (BZ) and *P. oceanica* (purple) in Italy (IT) were the dominant species at both sampling sites in this study. **b,** Box-and-whisker plots with overlaying data points show that pore waters collected underneath seagrass meadows and mangroves varied, depending on the type of meadow. Centre line is the median, box limits represent the upper and lower quartiles, whiskers are 1.5× the interquartile ranges, outliers are not shown. Specifically, sucrose was significantly higher in pore waters underneath the mixed ($n = 14$) and mono stands of the seagrass species *S. filiforme* ($n = 29$) and *T. testudinum* ($n = 47$), followed by samples taken underneath the mangrove peat ($n = 48$) and finally were lowest underneath *P. oceanica* meadows ($n = 382$). In almost all cases, excluding the samples from the mangroves, sucrose concentrations sometimes exceeded micromolar values. Post-hoc tests results: \*\*\*$P < 0.001$, \*\*$P < 0.01$. Contrasts not shown are not significant (Supplementary Tables 2 and 3).

centimetres in aquatic sediments[44,45]. We measured similarly high concentrations of sucrose as those in *P. oceanica* meadows in the pore waters of other marine plant ecosystems: in two other seagrass species from phylogenetically distinct lineages of angiosperms (*Thalassia testudinum* and *Syringodium filiforme)* and in a mixed stand of mangrove species (primarily *Rhizophora mangle*) (Fig. 6 and Supplementary Table 1). It is intriguing that sucrose persists in these otherwise oligotrophic ecosystems. To fully understand how these labile components of the DOC pool integrate into global carbon models, it is critical that future studies quantify the composition of this otherwise unrecognized carbon pool.

## Conclusions

Our study revealed that large amounts of the sucrose produced by *P. oceanica* did not fully enter the microbial loop, which probably has significant consequences for carbon burial in these coastal ecosystems. The accumulation of sucrose is driven by the presence of phenolics under hypoxic to anoxic conditions, which we show limited microbial metabolism of sugars. These results provide evidence that the reactivity of DOM is complex and governed by biotic and abiotic processes, including the metabolic potential of the microbial community, the availability of electron acceptors and inhibitors of microbial metabolism.

As a valuable energy resource, why does *P. oceanica* deposit sucrose to their sediments? *P. oceanica* release of sugars is probably an evolutionary mechanism to deal with overflow metabolism during periods of high productivity. Seagrasses evolved a reduced capacity to produce structural carbohydrates, such as cellulose, in favour of maintaining a large osmolyte pool of non-structural carbohydrates, such as sugars, to compensate for high salinity compared to their terrestrial relatives[46]. Seagrass release of excess photosynthate to their sediments in the form of sucrose could also serve as an overflow valve when productive rates exceed storage capacity or when nitrogen and phosphorus limit growth. In most marine oxygen-limited environments, plant release of sugars would stimulate sulfate reduction by sediment microbes leading to the accumulation of high concentrations of sulfide, which is toxic to seagrass roots[47–49]. However, we show that the presence of phenolics limits the microbial respiration of sugars under these low oxygen conditions. This would have the selective advantage that seagrasses could deposit their excess photosynthates to their sediments, while limiting the formation of sulfide.

Another selective advantage for *P. oceanica* in releasing sucrose is that it probably attracts key microbial partners that benefit plant fitness within the seagrass rhizosphere. In terrestrial plants, sucrose makes up 50% of their exudates during their early developmental stages. The early priming of the plant's rhizosphere with sugar is critical in shaping plant-associated microbial communities[35]. Plant exudation of sugars and other compounds is tied to chemotaxis of potential symbionts, including soil bacteria and mycorrhizal fungi[42]. Here, we show that in the seagrass rhizosphere, the vast majority of sediment microorganisms were not able to metabolize sucrose. However, we did identify a select set of putative sucrose specialists that have the metabolic potential to degrade phenolics and consume sucrose. Some of these taxa are also capable of sulfur and hydrogen oxidation and, intriguingly, nitrogen fixation. In oligotrophic waters, such as the Mediterranean, nitrogen is a limited resource for growth. Sucrose release by seagrasses, including *P. oceanica*, to their rhizosphere could support beneficial symbionts that metabolize sucrose and fix nitrogen in the presence of phenolics and absence of oxygen.

Future works needs to define the composition and decomposition rates of the DOC pool underneath aquatic plants and determine the lability of DOC as well as the proportion of buried carbon in seagrass and mangrove environments. This includes quantifying the rates at which aquatic plants excrete sucrose and other sugars to their rhizosphere using stable isotope probing to track the production and subsequent release of labelled metabolites[16,50]. Given that many plants excrete sugars and phenolics to their soils, and that in most aquatic environments sediments are largely anoxic, we predict that future studies will discover high concentrations of sugars in other rhizosphere environments. In addition to seagrasses and mangroves, this will include the rhizosphere of rice, marshlands and swamps. This study lays the foundation for future research to explore the processes that promote the burial and persistence of normally labile carbon and expand our understanding of carbon cycling in aquatic rhizospheres.

## Methods

**Pore water sampling.** Pore water was collected from vegetated and non-vegetated sediments for metabolomic analyses from *P. oceanica* seagrass meadows in the Mediterranean off the island of Elba, Italy (42° 48′ 29.4588″ N; 10° 8′ 34.4436″ E), the Caribbean at Carrie Bow Cay, Belize (N 16° 04′ 59″; W 88° 04′ 55″) and Twin Cayes, Belize (N 16° 50′ 3″; W 88° 6′ 23″) and the Baltic Sea off the coast of Kiel, Germany (54° 27′ 26.56256″ N; 10° 11′ 33.1908″ E). All water samples were collected using a blunt steel lance. For details regarding the experimental design for both spatial and temporal sampling of pore water and seagrass tissues see Supplementary Methods.

**Gas chromatography–mass spectrometry-based metabolomics.** Metabolomic profiles were obtained from sediment pore waters and prepared for gas

chromatography–mass spectrometry (GC–MS) using a recently described method[51]. This method targets the quantification and identification of lower weight molecular compounds. It is possible that there are other higher molecular weight metabolites, such as oligosaccharides and polysaccharides, dissolved in the pore water that were not captured by our analytical approach in the pore water samples. Briefly, 0.5 ml of individual pore water samples were dried to completeness in a speed-vacuum concentrator (~8 h). To further remove water from the salt pellet, we added 250 μl of toluene to the dried pellet and sonicated the mixture for 10 min. The toluene was subsequently removed under a steady flow of $N_2$ gas. The resulting extracts were stored until GC–MS analysis at 4 °C.

*Sample derivatization.* To remove condensation formed during extract storage, directly before preparation for GC–MS analysis we further dried extracts in a vacuum concentrator for 30 min. Metabolite derivatization was performed by adding 80 μl of methoxyamine hydrochloride dissolved in pyridine (20 mg ml⁻¹) to the dried pellet and incubating for 90 min at 37 °C using a thermal rotating incubator under constant rotation at 1,350 r.p.m. Following recent advancements in signal improvement[52], the pyridine was removed from the sample at room temperature under a gentle flow of $N_2$ gas (~1 h). Following the addition of 100 μl of *N*,*O*-bis(trimethylsilyl)trifluoroacetamide, each extract was vortexed and incubated for another 30 min at 37 °C using a thermal rotating incubator under constant rotation at 1,350 r.p.m. A total of 100 μl was transferred to a GC–MS vial for further GC–MS data acquisition.

*GC–MS data acquisition.* To obtain metabolite profiles, all derivatized samples were analysed on an Agilent 7890B GC coupled to an Agilent 5977 A single quadrupole mass selective detector. The injector temperature was set at 290 °C. Using an Agilent 7693 autosampler, 1 μl was injected in splitless mode through a GC inlet liner (ultra inert, splitless, single taper, glass wool, Agilent) onto a DB-5MS column (30 m × 0.25 mm, film thickness 0.25 μm; including 10 m DuraGuard column, Agilent). Metabolite separation on the column was achieved with an initial oven temperature of 60 °C followed by a ramp of 20 °C min⁻¹ until 325 °C was reached, which was then held for 2 min. Helium carrier gas was used at a constant flow rate of 1 ml min⁻¹. Mass spectra were acquired in electron ionization mode at 70 eV across the mass range of 50–600 *m/z* and a scan rate of two scans per second. The retention time was locked using standard mixture of fatty acid methyl esters (Sigma-Aldrich).

*Metabolome data analysis.* To identify differences in metabolites between vegetated and non-vegetated sediments, we used a non-targeted metabolomic approach to assess pore water samples collected inside and outside a *P. oceanica* seagrass meadow. Raw GC–MS data from samples collected in October 2016 were imported into R (v.3.5.2) as converted mzXML files (MsConvert)[53] and processed using XCMS (v.2.99.6)[54]. For further details on GC–MS data processing, see Supplementary Methods.

Following data processing, the resulting ion abundances from individual peaks were normalized to the ribitol internal standard and compared using a volcano plot to show ions with significant differences (alpha < 0.1) between sampling location and high fold change values (log(FC) > 2). These peaks were subsequently identified using the Mass Hunter Suite through comparison to the NIST database. The metabolite ID was confirmed by comparing peak retention times and mass spectra to authentic standards run on the same instrument.

Using our non-targeted analysis as a guide, identified sugars were further quantified across metabolomic profiles using Mass Hunter Quantification Suite (Agilent). Salinity matched calibration curves were used to determine absolute sugar concentrations in pore water samples using the same derivative across samples for each compound. Values below the detection limits were considered noise and given a value of 0 μM. The relative ion abundances of plant sugars were normalized to tissue wet weight and compared using an analysis of variance (ANOVA) across sampling time. Sugar concentrations from sediments were compared across spatial and temporal scales using ANOVA models where sugar concentration was a function of (1) sampling location ×sampling depth, (2) sampling month and (3) time of day blocked by sampling spot. Sugar concentrations were transformed to meet the assumptions of the model.

**Estimation of global sucrose abundances.** To determine the amount of sucrose underneath seagrass meadows in sediment pore waters, we first calculated the total volume of pore water underneath meadows worldwide for the estimated range in known area of seagrasses (300,000–600,000 km²) (ref. [3]). For both the estimated lower and upper surface areas of seagrass, we calculated the total volume of pore water using the following equation:

$$\text{l of pore water} = \text{median porosity} \times \text{area of seagrasses } (\text{km}^2)$$
$$\times \text{ sediment depth (km)}$$

where we used the value of 0.661 for median porosity of sediments worldwide[3] and a sediment depth of 0.0003 km for the measured distribution of sucrose based on our measurement in this study. The calculated minimum and maximum l of pore water were subsequently used to determine the total grams of sucrose using the following equation:

$$\text{grams of sucrose} = [\text{mean sucrose}] \times \text{ total l of pore water}$$

The mean sucrose concentration ($35.4\,\mu M$) was calculated by taking the average of measured values above the detection limits ($>0.02\,\mu M$) from sediment pore waters underneath all seagrasses in this study.

**DOC sampling and analysis.** DOC samples were collected in parallel with pore water metabolomic samples from inside, at the edge and 20 m outside a *P. oceanica* seagrass meadow in October of 2016. Taking care to avoid oversampling pore water space, ~20 ml of each pore water sample was collected into polypropylene syringes. Samples were filtered through precombusted (500 °C, 4 h) Whatman GF/F filters (0.7 µm) into 20 ml acid-washed and precombusted scintillation vials. Samples were acidified to pH 2 using 25% hydrochloric acid and stored at 4 °C until analysis. DOC concentrations were measured by high-temperature catalytic oxidation on a Shimadzu TOC-V$_{CPH}$ analyser. Instrument trueness and precision were tested against Deep Atlantic seawater reference material ($42\,\mu M$ DOC; acquired from the Hansell laboratory at University of Miami, FL, USA) and low carbon water ($1\,\mu M$ DOC) and was better than 5%.

**Molecular DOM analysis.** DOM samples were collected in parallel with pore water DOC samples from inside, at the edge and 20 m outside a *P. oceanica* seagrass meadow in October of 2016. DOM was extracted from filtered (0.7 µm GF/F) and acidified (pH 2) pore water samples using solid-phase extraction (SPE)[24]. For additional details regarding DOM analysis see Supplementary Methods.

**Seagrass root MS imaging.** Seagrass root pieces were embedded into precooled (4 °C) 2% carboxymethylcellulose blocks and snap frozen in liquid nitrogen. Roots were cross-sectioned into 12 µm thick slices with a cryrostat (Leica CM3050 S, Leica Biosystems Nussloch; chamber and object holder temperature at −25 °C) and either super 2,5-dihydroxybenzoic acid solution (S-DBH; $30\,mg\,ml^{-1}$ in 50% methanol:water with 0.1% trifluoroacetic acid (TFA)) or 9-aminoacridine matrix (9AA) in 70% methanol:water with 0.1% TFA was applied to individual sections using an automated pneumatic matrix sprayer (SunCollect, SunChrome, Wissenschaftliche Geräte). For additional sample preparation details see Supplementary Methods.

MS imaging was performed with an atmospheric pressure matrix-assisted laser desorption/ionization (AP-MALDI) ion source (AP-SMALDI10, TransMIT), coupled to a Fourier transform orbital trapping mass spectrometer (Q Exactive HF, Thermo Fisher Scientific). MS images were collected by scanning the matrix-covered tissue sections at a step size of 10 µm. Mass spectra were acquired in positive-ion mode for all S-DBH-prepared sections with a detection range of $m/z$ 50–750 and a mass resolving power of 240,000 at $m/z$ 200. Sections prepared with 9AA matrix were acquired in negative-ion mode with $m/z$ range 70–1,000 at a 20 µm step size.

**Oxygen detection.** Oxygen profiling within the seagrass bed was carried out in situ using a combination of custom-built and commercially available instruments on 30 August 2019 (Supplementary Methods gives a description of the instrument and procedure). Scuba divers collected two pore water profiles down to a depth of 30 cm in 2.5 cm intervals.

**Microbial activity.** Sediments were collected using cores inside ($n=3$) and at the edge ($n=3$) of a *P. oceanica* meadow in Sant'Andrea Bay, Elba, Italy on 27 August 2019 (42° 48′ 29.4588″ N; 10° 8′ 34.4436″ E; 6–8 m water depth). Individual cores were subsampled inside a glove bag under nitrogen gas to avoid introducing oxygen to the anoxic sediment layers. Cores were split into oxic (upper 3 cm) and anoxic fractions (5–10 cm). Individual fractions were mixed and 10 ml of sediment volume were added to each serum vial. Replicate 250 ml serum vials for each fraction were filled with filtered seawater (environmental control), filtered artificial seawater (technical control) or filtered artificial seawater with ~35 µM of phenolic extract prepared from seagrass tissue as previously described (Supplementary Methods)[55].

We introduced an equal amount of phenolics and sugars as this ratio was reported for other seagrass leachates[30]. Assuming the main phenolic compound is chicoric acid (Supplementary Fig. 6), this corresponds to a concentration of ~35 µM. All serum vials had 100 ml of headspace with either N$_2$ gas (anoxic) or air (oxic). To remove background levels of phenolics and other compounds, sediments incubated with artificial seawater were flushed with five times the sediment volume in artificial seawater before transfer to serum vials. Flushing with artificial seawater may have inadvertently removed additional cells, nutrients or metals that could have lowered metabolic rates compared to incubations with natural seawater. Fully labelled $^{13}C_{12}$-sucrose was added to each bottle at a final concentration of 50 µM. Oxic incubations were prepared in ambient air, with optode spots confirming the presence of oxygen throughout the experiment. Anoxic incubations under nitrogen atmosphere and both natural and artificial seawater were bubbled with N$_2$ gas until $<2\,\mu M$ of oxygen was measured.

Incubation seawater from each serum bottle was collected for GC–MS-based metabolomics and cavity ring-down spectrometry (CRDS). At each time point, 5 ml of incubation seawater were collected from each vial and replaced by either filtered air (oxic) or N$_2$ gas (anoxic). A total of 2 ml was transferred to an Eppendorf tube and stored at −20 °C until analysis (GC–MS), while the remaining 3 ml were transferred to a nitrogen-filled exetainer with 50 µl of saturated mercury(II) chloride solution to halt microbial metabolism. All samples where evidence indicated microbial metabolism was not successfully halted (cross-validated using metabolomics results) were excluded from the CRDS analysis.

The metabolic activity of the sediment was assessed by monitoring the disappearance of $^{13}C_{12}$-sucrose (GC–MS) and the production of $^{13}CO_2$ (CRDS) over time. Samples were prepared for GC–MS following the metabolomics protocol for seawater described above. To determine the rate of $^{13}CO_2$ production from $^{13}C_{12}$-sucrose, 3 ml of each sample were acidified with 50 µl of 20% phosphorphic acid in preparation for cavity ring-down spectroscopy (G2201-i coupled to a Liaison A0301, Picarro, connected to an AutoMate Prep Device, Bushnell)[56]. The production of $^{13}CO_2$ from $^{13}C$-sucrose was calculated by subtracting the initial $^{13}CO_2$ concentration from each subsequent time point for each bottle. The production of $^{13}CO_2$ from $^{13}C_{12}$-sucrose was calculated from the increase of $^{13}CO_2$ over 24 h. The values were corrected by subtracting the $^{12}CO_2$ background multiplied by the natural abundance of $^{13}C$.

Volumetric sucrose oxidation rates were determined by linear regression of the measured $^{13}C_{12}$-sucrose and $^{13}CO_2$ concentrations. The rates from the incubations were corrected for dilution and sediment porosity assumed as 0.42 (on the basis of previous calculations). A correction based on the measured $^{12}C_{12}$-sucrose concentration was applied to account for dilution of the labelled substrate. An active layer thickness of 3 cm for oxic and 20 cm for anoxic was assumed on the basis of the average sucrose concentrations in the sediment and multiplied with the volumetric rates (Supplementary Table 5 gives rate calculations).

**Detection of phenolic compounds from pore water.** We verified the correct identification of the phenolic compound caffeic acid (sum formula $C_9H_8O_4$) in our ultrahigh-resolution MS analyses of pore water DOM, using liquid chromatography–mass spectrometry (LC–MS) (Supplementary Fig. 7). About 0.5 ml of pore water were collected from underneath a seagrass meadow in July 2016 and processed as described above in 'Molecular DOM analysis'. After eluting samples from the SPE cartridge with methanol, samples were dried under vacuum for 8 h at 30 °C using the V-AL mode. Resulting extracts were redissolved in 500 µl of methanol, dried down in LC–MS Ultra High Recovery vials, redissolved in 50 µl of 10% methanol:90% MQ H$_2$O and analysed directly by LC–MS (Supplementary Methods).

**Sediment nucleic acid extraction, sequencing and analysis.** Genomic DNA and total RNA were extracted from sediment cores ($n=3$) collected inside, at the edge and 20 m outside a *P. oceanica* meadow in Sant'Andrea Bay, Elba, Italy, in October 2016. We explored the microbial communities at 10–15 cm because this sediment depth contained the highest sucrose concentrations in pore water profiles and was the average depth of *P. oceanica* root penetration. Directly after collection, cores were sectioned into 5 cm slices and frozen at −20 °C. A subsample of each sediment slice was also preseved in RNAlater (Sigma-Aldrich) for RNA extraction. Samples were stored at −80 °C until further processing.

*Nucleic acid extraction.* DNA was extracted from 0.2 g of the 10–15 cm slice. Before extraction, extracellular DNA was removed from the sediments following previously described methods[57,58]. Total RNA was extracted from a separate 0.2 g aliquot of the 10–15 cm slice of each sediment core preserved in RNAlater following a modified version of previously described methods for RNA recovery from marine sediments[59]. For additional details of DNA and RNA extraction, see Supplementary Methods.

*Library preparation and sequencing.* Library preparation and sequencing was performed at the Max Planck Genome Center Cologne, Germany (https://mpgc.mpipz.mpg.de/home/) (Supplementary Methods).

*Taxonomic composition using 16S amplicon reads.* Resulting 16S CCS reads from our PacBio sequing approach were processed using Dada2 (ref. [60]) in R to obtain ASVs. Briefly, amplicon reads were imported into R and filtered to control for expected sequence length (1,000–1,600 base pairs) and high-quality reads (minQ = 3). Following error estimation, taxonomy was assigned to each unique ASV by comparison against the GTDB database. The resulting count table was imported into phyloSeq[61] for subsequent analyses including estimate of sequence richness and comparison of taxonomic profiles across samples.

*Taxonomic composition from metagenomic libraries.* The pipeline phyloFlash (v.2.0)[62] was used to assemble the small subunit of the rRNA gene from each metagenomic library to assess taxonomic composition. The resulting sequence counts, taxonomically classified at order level through comparison to the SILVA SSU 111 (July 2012)[63] database, were imported into R (v.3.5.2) and processed using the phyloseq package (v.1.26.1)[61] to facilitate analysis and visualization. Data processing included removal of eukaryotic taxa and normalization of count data to relative abundances. A PCoA was used to assess differences in taxonomic compositions among sampling sites.

*Metagenomic bin recovery.* Metagenomic libraries were used to recover metagenome assembled genomes (MAGs) from marine sediments. Using the bbtools suite (v.38.34, http://jgi.doe.gov/data-and-tools/bb-tools/) reads were cleaned of Illumina adaptors and low-quality ends were trimmed before normalization to a maximum k-mer depth of 100X (k-mer length of 31). Reads with an average k-mer depth <2 were considered erroneous and were discarded. SPAdes (v.3.12.0)[64] was used to error-correct all remaining reads, which were then fed into MEGAHIT (v.1.1.3)[65] to generate a co-assembly with k-mers from 21 to 51 in steps of 10. Sequencing coverage of each contig was calculated by mapping the error-corrected reads back to the co-assembly using bbmap (v.38.34, http://jgi.doe.gov/data-and-tools/bb-tools/).

The metagenomic co-assembly was binned using three automated binning programs (MetaBAT v.0.32.5, concoct v.1.0.0 and MaxBin v.2.2.6)[66–68] and subsequently dereplicated and aggregated with DAS Tool (v.1.1.1)[69]. Bin quality was assessed using the CheckM lineage-based work flow method (v.1.0.7) (Supplementary Table 7)[70]. Bins with completeness ≥50% and contamination scores corrected for strain heterogeneity ≤10% were considered to be of medium draft quality and used in subsequent analyses. Resulting MAGs were taxonomically classified using the GTDBTk classify work flow (v.0.2.1)[71]. MAGs represented only 2.9% of the metagenomic read set; however, their taxonomic identities reflected the relative abundance of the sediment communities occurring across habitats (Supplementary Table 7 and Extended Data Fig. 5). Open reading frames from each MAG were predicted with Prodigal (v.2.6.4)[72] and annotated using a combination of reference database searchers: dbCAN (v.2.0)[73], diamond blastp (v.0.9.25)[74], Uniprot[75], interproscan (v.5.36-75)[76] against both PFam[77] and Phobius[78] and EggNOG mapper (v.2.0)[79].

*Expression analysis.* Metatranscriptomic reads were cleaned of Illumina adaptors and low-quality ends were trimmed using the bbtools suite (v.38.34, http://jgi.doe.gov/data-and-tools/bb-tools/) before error correction using SPAdes (v.3.12.0)[64]. Ribosomal RNA was filtered from cleaned reads using SortMeRNA (v.2.1b)[80]. Non-rRNA reads were mapped to all bins and counted in Kallisto (v.0.46)[81]. Raw read counts were imported into R (v.3.5.2) for analysis and visualization. To correct for varying relative abundances in MAGs across samples, transcript counts were normalized to one million per each MAG in each sample (transcripts per million, TPM). TPM values for genes with identical annotation were summed within samples. Total transcript counts per habitat per MAG were used to compare transcription levels inside, at the edge and outside the meadow.

*Phylogenomic tree construction.* Phylogenomic trees were constructed from select MAGs using GToTree (v.1.4.14)[82]. Briefly, genomes related to each MAG (on the basis of GTDBtk taxonomy) and at least three outgroups were downloaded from NCBI RefSeq or Genbank repositories. Genes were called using Prodigal[72] and annotated through HHMER3 (ref. [83]) comparisons to a set of 74 bacterial single-copy genes. Genomes were retained in the analysis if they contained at least 45% of all single-copy genes. Annotated genes were aligned using MUSCLE[84] and trimmed with TrimAl[85]. IQ-TREE[86] was used to calculate a maximum-likelihood tree with ultrafast bootstrap support values from the concatenated alignment which was visualized using the Interactive Tree of Life webserver[87].

**Reporting Summary.** Further information on research design is available in the Nature Research Reporting Summary linked to this article.

## Data availability
Sequence data from this study were deposited in the European Nucleotide Archive under accession numbers PRJEB35096 and PRJEB40297 using the data brokerage service from the German Federation for Biological Data[88], in compliance with the Minimal Information about any (X) Sequence (MIxS) standard[89]. Metabolomics data were deposited in Metabolights (https://www.ebi.ac.uk/metabolights/) under accession numbers MTBLS1570, MTBLS1610, MTBLS1579 and MTBLS1746. All other datasets are available at https://zenodo.org/record/3843378.

## Code availability
All code used for data processing and analysis are available at https://github.com/esogin/sweet_spots_in_the_sea.

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

## Acknowledgements

We thank T. Gulstad, K. Caspersen, M. Weinhold, F. Fojt, J. Beckmann, S. Wetzel and M. Sadowski (Max Planck Institute for Marine Microbiology, MPI-MM) for support with data acquisition and sample preparation. We also thank B. Huettel (Max Planck Genome Center) for his support with sequencing. We are grateful for fruitful discussions with colleagues at the MPI-MM, particularly M. Kuypers, D. de Beer, D. Koopmans, as well as many other colleagues in the Departments of Symbiosis and Biogeochemistry, and T. Dittmar (University Oldenburg). We thank M. Weber, C. Lott and HYDRA staff for sample collections. We are grateful to the Max Planck Society, the Gordon and Betty Moore Foundation (Marine Microbial Initiative Investigator Award to N.D., grant no. GBMF3811) and the MARUM Cluster of Excellence 'The Ocean Floor' (Deutsche Forschungsgemeinschaft (German Research Foundation) under Germany's Excellence Strategy EXC-2077-390741603) for financial support. This work is contribution no. 1059 from the Carrie Bow Cay Laboratory, Caribbean Coral Reef Ecosystem Program, National Museum of Natural History, Washington DC.

## Author contributions

E.M.S., N.D. and M.L. conceived the study. E.M.S. and D.M. collected, processed and analysed primary samples for DOM, metabolomics and sequencing analyses. E.M.S. performed statistical analysis for all primary data. M.S. analysed samples for DOM analysis. S.A. collected and analysed samples for dissolved oxygen concentrations. B.G. collected and analysed MS imaging data. P.B. performed sediment incubation experiments and P.B., S.A. and S.S. analysed data from these incubations. E.M.S., H.G.-V. and D.V.M. designed and optimized bioinformatic pipelines for sequencing analysis. D.V.M. and G.D. reconstructed microbial metabolisms on the basis of sequencing results. G.P. contributed to data interpretation and genomic analyses. E.M.S. wrote the manuscript together with M.L. and N.D., with all other authors contributing to the writing, editing and review of the manuscript.

## Funding

## Competing interests

The authors declare no competing interests.

## Additional information

**Extended data** is available for this paper at https://doi.org/10.1038/s41559-022-01740-z.

**Correspondence and requests for materials** should be addressed to E. Maggie Sogin, Nicole Dubilier or Manuel Liebeke.

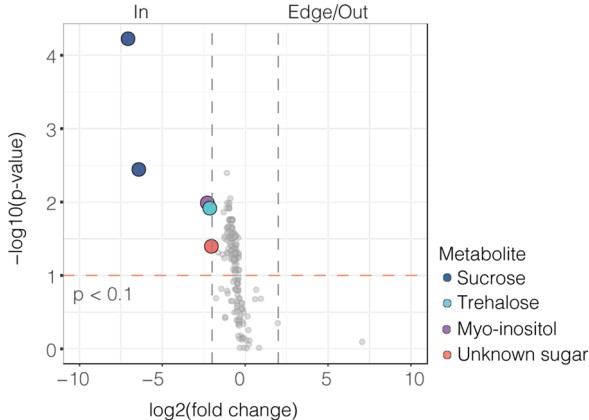

**Extended Data Fig. 1 | Sugars were more abundant in pore waters underneath *P. oceanica* than in unvegetated sediments.** A volcano plot comparing gas chromatography–mass spectrometry peaks from pore water metabolites collected inside (In) versus 1 and 20 m away from a seagrass meadow (Edge/Out). Significant peaks (Benjamini-Hochberg adjusted one-way ANOVA *p-value* < 0.1) are represented by coloured circles (α < 0.1). The grey dashed lines represent Log2-fold changes > 2; The orange dashed line represents Benjamini-Hochberg corrected *p-values* < 0.1.

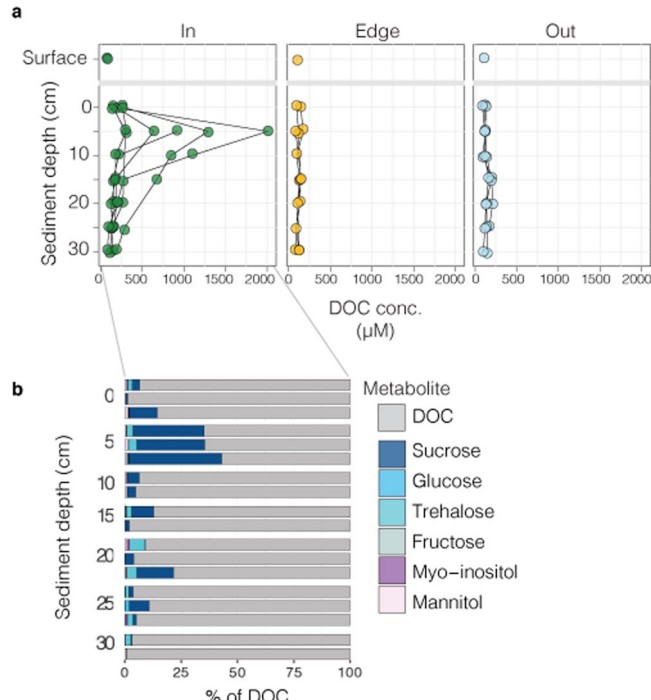

**Extended Data Fig. 2 | Dissolved organic carbon (DOC) concentrations were higher inside the meadow then at the edge or outside. a**, DOC concentrations (Supplementary Table 4) in sediment pore water profiles from inside ($n=6$), at the edge ($n=3$) and outside ($n=3$) of a *P. oceanica* meadow show DOC concentrations were significantly higher in pore waters inside the meadow and varied as a function of sediment depth (two-way ANOVA $p<0.001$; Supplementary Table 2 and Supplementary Table 3). **b**, A random subset ($n=3$) of DOC samples collected inside the meadow show that sugars made up to 40% of the DOC composition within the 5 cm depth, where seagrass roots dominated the sediment.

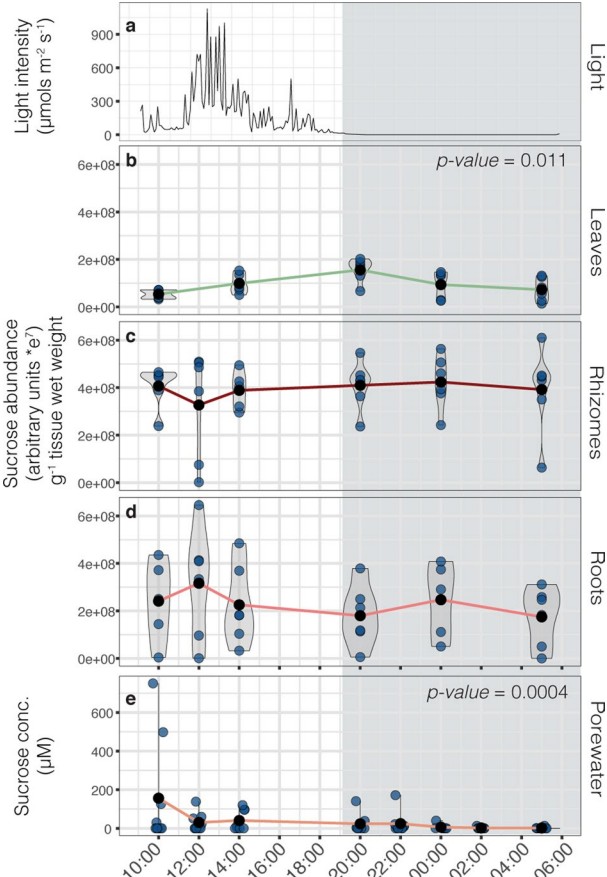

**Extended Data Fig. 3 | Sucrose abundances varied across a day/night cycle in plant leaves and sediment pore water. a**, In situ light levels across 24 hours in a Mediterranean *P. oceanica* meadow off the Island of Elba, Italy (> 2 m water depth). **a-d**, Violin plots show the relative abundance of sucrose g⁻¹ tissue plant wet weight in **b**, leaves, **c**, rhizomes, and **d**, roots at each sampling time point. Linear models show that sucrose relative abundances were significantly higher in plant leaves at dusk (t = 20:00) than at other times. **e**, Violin plots showing sucrose concentrations measured in sediment pore waters at each sampling time point. A linear mixed effects model with a random effect of sampling spot shows sucrose concentrations significantly changed throughout the day and were higher during the daylight hours then after midnight (t = 0:00) and before dawn (t = 5:00). The means of each sampling time point (black points) are connected by solid lines. For **e**, sucrose concentrations greater than the limits of quantification are represented as 200 µM. All statistical results are reported in Supplementary Table 2 and Supplementary Table 3. All data points were transformed before statistical analyses to meet assumptions of normality.

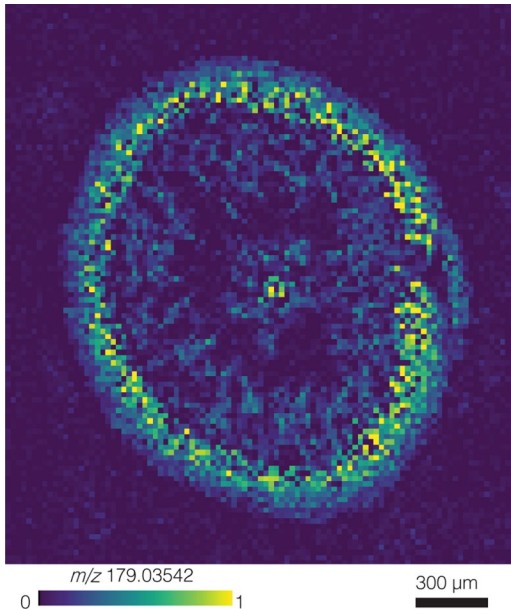

**Extended Data Fig. 4 | Seagrass roots contained phenolic compounds.** Matrix-assisted laser desorption/ionization mass spectrometry image from a Mediterranean *P. oceanica* root cross-section collected off the Island of Elba, Italy. The ion image shows the presence of the phenolic compound, identified as caffeic acid ($C_9H_8O_4$, [M-H]$^-$, *m/z* 179.0354; Supplementary Fig. 7). Relative ion abundances are visualized from low (blue) to high (yellow).

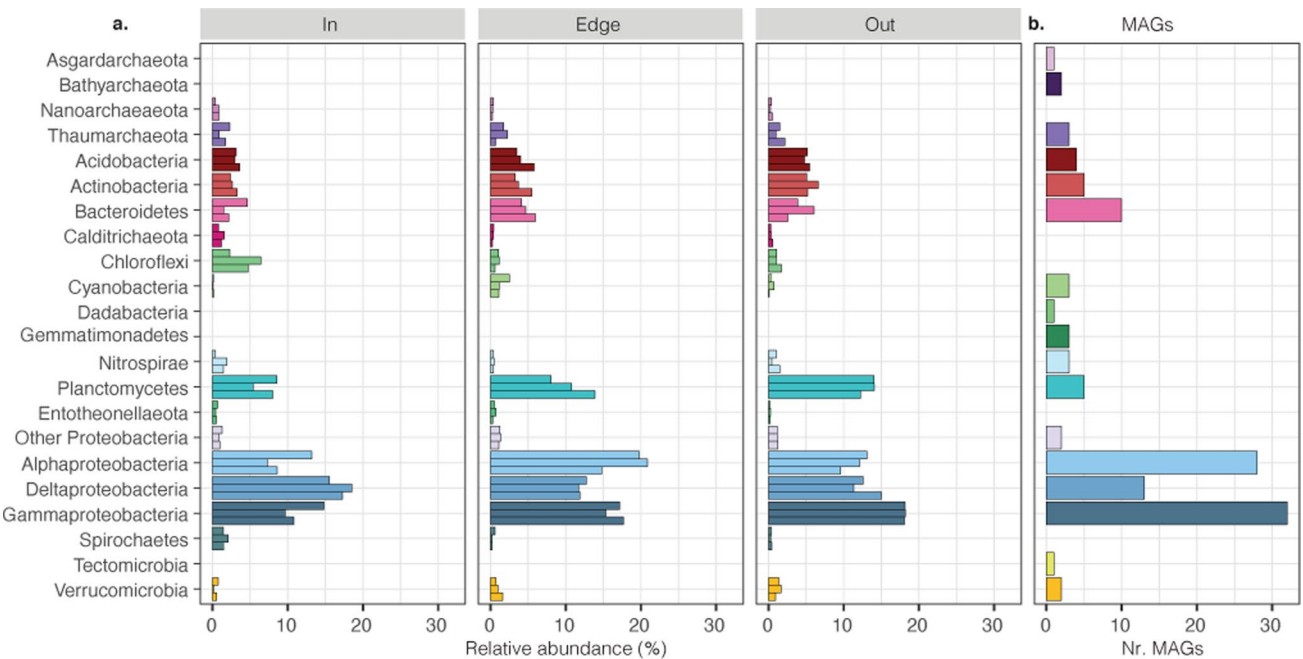

**Extended Data Fig. 5 | Composition of sediment microbial communities according to phyla. a**. The relative abundance of taxonomic phyla from each metagenomic library collected inside, at the edge and outside a Mediterranean *P. oceanica* meadow off the island of Elba, Italy. **b**. The number of reconstructed MAGs classified by phylum, obtained from the binned co-assembly of all metagenomic reads from these sediments.

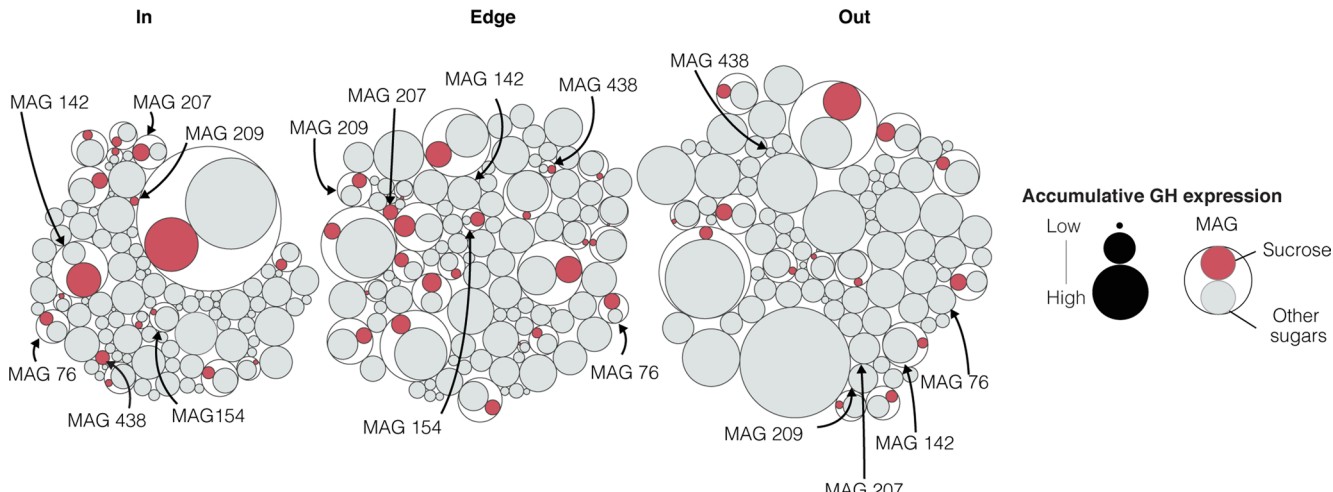

**Extended Data Fig. 6 | Sucrose specialists showed higher expression of genes for metabolizing sucrose than other sugars.** Circle packing plots show the hierarchical relationships between the accumulative expression of glycoside hydrolases (GH) for each MAG from underneath (In), 1m (Edge) and 20 m (Out) away from a *P. oceanica* meadow off Elba, Italy. Inner circles are coloured according to the predicted substrates (red = sucrose, light grey = other sugars). The size of the inner circle represents the accumulative transcription level for each MAG (outer white circles) across collection sites. Sucrose specialists are labelled with their MAG numbers.

# Reporting Summary

## Statistics

For all statistical analyses, confirm that the following items are present in the figure legend, table legend, main text, or Methods section.

| n/a | Confirmed | |
|---|---|---|
| ☐ | ☒ | The exact sample size (*n*) for each experimental group/condition, given as a discrete number and unit of measurement |
| ☐ | ☒ | A statement on whether measurements were taken from distinct samples or whether the same sample was measured repeatedly |
| ☐ | ☒ | The statistical test(s) used AND whether they are one- or two-sided<br>*Only common tests should be described solely by name; describe more complex techniques in the Methods section.* |
| ☒ | ☐ | A description of all covariates tested |
| ☐ | ☒ | A description of any assumptions or corrections, such as tests of normality and adjustment for multiple comparisons |
| ☐ | ☒ | A full description of the statistical parameters including central tendency (e.g. means) or other basic estimates (e.g. regression coefficient) AND variation (e.g. standard deviation) or associated estimates of uncertainty (e.g. confidence intervals) |
| ☒ | ☐ | For null hypothesis testing, the test statistic (e.g. *F*, *t*, *r*) with confidence intervals, effect sizes, degrees of freedom and *P* value noted<br>*Give P values as exact values whenever suitable.* |
| ☒ | ☐ | For Bayesian analysis, information on the choice of priors and Markov chain Monte Carlo settings |
| ☒ | ☐ | For hierarchical and complex designs, identification of the appropriate level for tests and full reporting of outcomes |
| ☒ | ☐ | Estimates of effect sizes (e.g. Cohen's *d*, Pearson's *r*), indicating how they were calculated |

*Our web collection on statistics for biologists contains articles on many of the points above.*

## Software and code

Policy information about availability of computer code

| | |
|---|---|
| Data collection | No software was used for data collection. |
| Data analysis | MsConvert<br>XCMS, v2.99.6<br>CAMERA, v1.32.0<br>Mass Hunter Suite<br>Scils 2019a<br>Dada2<br>phyloFlash, v2.0, https://github.com/HRGV/phyloFlash<br>phyloseq, v1.26.1<br>BBmap suite, v38.34, sourceforge.net/projects/bbmap<br>SPAdes, v3.12<br>MEGAHIT, v1.13<br>MetaBAT, v0.32.5<br>concoct, v1.0.0<br>MaxBin, v2.2.6<br>DAS Tool, v1.1.1<br>CheckM, v1.0.7<br>GTDBTk, v0.2.1<br>Prodigal, v2.6.4<br>dbCAN, v2.0,<br>diamond blastp, v0.9.25<br>interproscan, v.5.36-75 |

EggNOG mapper, v2.0
SortMeRNA, v2.1b
Kallisto, v0.46
GToTree, v1.4.14
HHMER3
MUSCLE
TrimAl
IQ-TREE

For manuscripts utilizing custom algorithms or software that are central to the research but not yet described in published literature, software must be made available to editors and reviewers. We strongly encourage code deposition in a community repository (e.g. GitHub). See the Nature Portfolio guidelines for submitting code & software for further information.

## Data

Policy information about availability of data

All manuscripts must include a data availability statement. This statement should provide the following information, where applicable:
  - Accession codes, unique identifiers, or web links for publicly available datasets
  - A description of any restrictions on data availability
  - For clinical datasets or third party data, please ensure that the statement adheres to our policy

Sequence data from this study were deposited in the European Nucleotide Archive under accession numbers PRJEB35096 and PRJEB40297 using the data brokerage service from the German Federation for Biological Data (GFBio), in compliance with the Minimal Information about any (X) Sequence (MIxS) standard. Metabolomics data were deposited in Metabolights (https://www.ebi.ac.uk/metabolights/) under accession numbers MTBLS1570, MTBLS1610, MTBLS1579, and MTBLS1746. All other datasets are available at zenodo.org under doi: 10.5281/zenodo.3843378.

# Field-specific reporting

Please select the one below that is the best fit for your research. If you are not sure, read the appropriate sections before making your selection.

☐ Life sciences        ☐ Behavioural & social sciences        ☒ Ecological, evolutionary & environmental sciences

For a reference copy of the document with all sections, see nature.com/documents/nr-reporting-summary-flat.pdf

# Ecological, evolutionary & environmental sciences study design

All studies must disclose on these points even when the disclosure is negative.

| Study description | Porewater profiles and sediment samples were collected to describe the microbial ecology of the seagrass rhizosphere. Due to the complexity of the associated datasets, please refer to Tables S1 and S2 for sample sizes and statistical approaches used in this study unless otherwise indicated below. |
|---|---|
| Research sample | Porewater metabolomics. Porewater samples were collected at multiple locations and multiple time points across a multi-year study exploring the metabolite composition underneath seagrass meadows. Depth profiles consisted of taking 2 mL of porewater every 5 cm from the sediment surface to -30 or -40 cm beneath the meadow. In Sant'Andrea Bay, Elba, Italy, porewater samples were collected underneath the meadow, 1 m and 20 m away from the meadow. Depending on sampling location and month sampled, 6-9 individual depth profiles were collected from each sampling site. Porewater samples from Galanzana Bay, Elba, Italy, were also collected across a 24 hour period using fixed lances to reduced variation based on sampling location: At each time point we collected between 6 and 10 samples, as some samples were lost due to clogging of the porewater lance at the time of sampling.<br><br>Dissolved organic carbon. 20 mL of porewater samples were also collected for dissolved organic carbon analysis from underneath, 1 m and 20 m away from a seagrass meadow in Sant'Andrea Bay, Elba, Italy (n=6). From each site, a subset of these samples (n=3) were also analyzed for dissolved organic matter composition.<br><br>Seagrass metabolomics. Replicate seagrass samples (n=6 per sampling time point) for metabolomic analysis were collected from Galanzana Bay, Elba, Italy over a 24 hour time period at the same time as the 24 hour samples collected for porewater metabolomics. Seagrass samples were used to measure bulk sucrose concentrations from the seagrass leaves, rhizomes and root tissues, as well as explore the distribution of sucrose within the seagrass root (n=8).<br><br>Sediment samples. Sediment samples were collected from Sant'Andrea Bay Elba, Italy for metagenomic and metatranscriptomic analysis. Samples were taken underneath the seagrass meadow, 1 m and 20 m away from the seagrass meadow (n=3 per site).<br><br>Metabolic activity experiments. Sediment cores were collected underneath and 1 m away from seagrass meadow from Sant'Andrea Bay, Elba, Italy (n=3). Sediments were incubated under oxic or anoxic conditions, and in the presence or absence of phenolic compounds. 13C-sucrose consumption was monitored over time. Sucrose respiration rates were calculated from these incubations, but not statistically compared. |
| Sampling strategy | Porewater samples were collected using a steel lance (1 m long, 2 µm inner diameter) outfitted with a wire mesh (63 µm) to prevent the intake of sediment and seagrass, porewater was slowly extracted from sediments into sterile syringes.<br><br>Individual seagrass plants were collected by hand and immediately immersed in liquid nitrogen to halt changes in metabolite composition during sampling. |

Sediment samples for metagenomic and metatranscriptomic analyses were collected using push cores. Directly after collection, cores were sectioned into 5 cm slices and frozen at -20 °C. A subsample of each sediment slice was also preserved in RNAlater (SigmaAldrich) for RNA extraction.

Dissolved organic carbon (DOC) and dissolved organic matter (DOM) samples were collected in parallel with porewater metabolomic samples from inside, at the edge and 20 m outside a P. oceanica seagrass meadow in October 2016. DOC/DOM samples were filtered through pre-combusted (500 °C, 4 h) Whatman GF/F filters (0.7 µm) into 20 mL acid-washed and pre-combusted scintillation vials. Samples were acidified to pH 2 using 25% hydrochloric acid and stored at 4 °C until analysis.

Samples for measuring the rate of sucrose consumption by seagrass sediments were frozen for metabolomic analysis at each sampling time point (0, 3, 6, 12 and 24 hours post 13C-sucrose introduction). Furthermore, samples for cavity ring-down spectroscopy to measure the production of 13CO2 were also collected at each time point by halting biological activity in 3 mL of each sample using 50 µl saturated HgCl2 solution. For each experimental condition 3 replicate incubation bottles were prepared from 3 different sampling cores inside and 1 m away from the seagrass meadow. .

For all samples collected, sample size was chosen according to sizes used in comparable studies.

| | |
|---|---|
| Data collection | All metabolomic data was collected using either gas chromatography-mass spectrometry or mass spectrometry imaging using instruments at the Max Planck Institute for Marine Microbiology by Dr. M. Sogin, Dr. B. Geier and D. Michellod. Dissolved organic carbon and dissolved organic matter samples were analyzed by Dr. M. Seidel at the University of Oldenberg. Dr. S. Ahmerkamp collected in situ oxygen concentrations from a seagrass meadow in Sant'Andra Bay, Italy. P. Bourceau and S. Schorn collected cavity ring-down spectrometer data from sediment incubation experiments. The gene expression data were generated by DNA and RNA sequencing at the Max Planck Genome Centre in Cologne under the supervision of Dr. B. Huettel. |
| Timing and spatial scale | Initial metabolomics porewater profiles were collected from Sant'Andrea Bay, Elba, Italy in April, July and October 2016. Further porewater samples and seagrass tissue samples were collected across a 24 hour period from Galanzana Bay, Elba, Italy in May 2017. Porewater samples collected underneath seagrass meadows off Twin Cayes and Carrie Bow Caye, Belize were collected in April 2017, and Kiel Bight, Germany in August 2018. All porewater profiles consisted of samples taken every 5 cm from the sediment surface to 30 to 40 cm below the sediment surface. Sediment samples for metagenomics and transcriptomics were collected in October 2016 in Sant'Andrea Bay. Metabolic activity experiments using seagrass sediments from Sant'Andrea Bay were performed in September 2019 with freshly collected sediment material. |
| Data exclusions | No data were excluded from the analysis. |
| Reproducibility | All attempts to repeat the experiments were successful. |
| Randomization | All samples were randomly collected from seagrass habitats without a priori expectations that the sampling would influence the analysis. |
| Blinding | Blinding was not preformed because it was not relevant to this study. This study was an exploratory investigation into the microbial ecology of seagrass meadows, without a priori expectations that would influence the analysis. |

Did the study involve field work?  ☒ Yes  ☐ No

# Field work, collection and transport

| | |
|---|---|
| Field conditions | Field work was conducted across a wide variety of field sites and time points during this multi-year study. |
| Location | Samples were collected at the following locations:<br>Sant'Andrea Bay, Elba, Italy (42° 48'29.4588" N; 10° 8' 34.4436" E ; 6-8 m water depth)<br>Galanzana Bay, Elba, Italy (42° 44'9.438" N; 10° 14' 16.3032" E; 2 m water depth)<br>Caribbean at Carrie Bow Cay, Belize (N 16° 04' 59"; W 88° 04' 55"; 2 m water depth)<br>Twin Cayes, Belize (N 16° 50' 3"; W 88° 6' 23", 2 m water depth)<br>Baltic Sea off the coast of Kiel, Germany (54° 27' 26.56256" N; 10° 11' 33.1908" E, 2 m water depth) |
| Access & import/export | All samples collected outside of the European Union were collected under the guidance of the Carrie Bow Cay Laboratory, Caribbean Coral Reef Ecosystem Program, National Museum of Natural History, Washington DC. These samples and samples collected from waters within the European Union were collected in compliance with local, national and international laws. |
| Disturbance | No disturbance was caused by the study. |

# Reporting for specific materials, systems and methods

We require information from authors about some types of materials, experimental systems and methods used in many studies. Here, indicate whether each material, system or method listed is relevant to your study. If you are not sure if a list item applies to your research, read the appropriate section before selecting a response.

## Materials & experimental systems

| n/a | Involved in the study |
|---|---|
| ☒ ☐ | Antibodies |
| ☒ ☐ | Eukaryotic cell lines |
| ☒ ☐ | Palaeontology and archaeology |
| ☒ ☐ | Animals and other organisms |
| ☒ ☐ | Human research participants |
| ☒ ☐ | Clinical data |
| ☒ ☐ | Dual use research of concern |

## Methods

| n/a | Involved in the study |
|---|---|
| ☒ ☐ | ChIP-seq |
| ☒ ☐ | Flow cytometry |
| ☒ ☐ | MRI-based neuroimaging |

