## [Peer Review file · Nature Ecology & Evolution]

Peer Review Information

Journal: Nature Ecology & Evolution

Manuscript Title: Sugars dominate the seagrass rhizosphere

Corresponding author name(s): E. Maggie Sogin, Nicole Dubilier, Manuel Liebeke

Editorial Notes:

Reviewer Comments & Decisions:

Decision Letter, initial version:
--

21st January 2022

Dear Dr. Sogin,

Thank you for submitting your revised manuscript "Sugars dominate the seagrass rhizosphere" (NATECOLEVOL-211215425-T). It has now been seen again by two of the previous reviewers from your submission to Nature, whose comments are below (please note that the numbering of the referees is the same as previously). The reviewers find that the paper has improved in revision, and therefore we'll be happy in principle to publish it in Nature Ecology & Evolution, pending minor revisions to satisfy the reviewers' final requests and to comply with our editorial and formatting guidelines.

In particular, Referee #2 has requested some additional statistical analysis to further strengthen the robustness of the study.

[REDACTED]

Reviewer #2 (Remarks to the Author):

This manuscript asserts that phenolics inhibit microbial sucrose metabolism within anoxic sediments under seagrass meadows leading to accumulation. This is an interesting and important topic. The manuscript is well-written with a compelling narrative and attractive figures. I have reviewed this manuscript previously and am very pleased to see that the authors have prepared a substantially improved manuscript in response to earlier reviews. A few issues remain as described below.

Figure S2 is nice and the same should be included for the other sugars in S1.

Figure S7 I'm not clear how you are differentiating caffeic vs. chicoric acids given that they appear to have the same retention time on your GC-MS method, you apparently see the same parent ion m/z 396 for both and in the caption you state they have similar fragmentation spectra. I would expect chicoric acid to elute after caffeic acid. Might it be the peak at 25min? What am I missing?

Figure S3b caption is unclear "A subset of the DOC samples collected inside the meadow show that sugars made up to 40% of the DOC composition within the 5 cm depth, where seagrass roots dominated the sediment."

This figure is very important to support the importance of sucrose to the overall global carbon stock (this could be in the main text). However, it is unclear what 'A subset of DOC samples' means? How were the samples selected?

I also couldn't find the DOC data. Figure S7 refers to table S2 but the tab/table labeled 'table s2' provides the ^{13}C incubation data.

Double check all of the figure and table references. For example line 177 DOM of *P. oceanica* pore waters was highest underneath the seagrass meadow between 5 and 25 cm sediment depth (Extended Data 4b; see Fig. S8 for identification caffeic acid in pore waters). Yet ED 4b shows the phenolics data. ED 3 fig3b shows DOC values.

Line 176 to 179: Did you perform statistical analysis to support the assertion that caffeic acid abundance between different depths? This is required to support this assertion.

Lines 312-316 and related text. There isn't direct evidence that these clades of microbes are using sucrose so statements like "The three sucrose specialists that used sucrose inside the meadow" are inferences. Suggest rewording to something along the lines of "based on the genomic analysis we predict that it is these three sucrose specialists that used sucrose...."

The 'n' number (number of replicates) is missing from most of the figures (e.g. Fig 1b,2a,2b,4b,S1,S3)

2You should perform hoc test for ANOVA is missing for example using hoc Tukey test for results showed in Fig 1b, 4b, S1, S3a, S3b,S4, S6.

Minor comments:

Currently it is very hard to find the SI tables. All supplementary tables should have the name of the table on the corresponding tab of the excel file. The use of both SI tables and figures as well as extended data makes this even more confusing. Is there a need to have both extended data and SI? For example "DOM of *P. oceanica* pore waters was highest underneath the seagrass meadow between 5 and 25 cm sediment depth (Extended Data 4b; see Fig. S8 for identification caffeic acid in pore waters).

Reviewer #3 (Remarks to the Author):

The authors have done a careful revision of the manuscript based on the comments from the reviewers. There were many questions/comments and I find that the manuscript now appears in a much more reflected version taken into account the uncertainties of the study. The results of the study are exciting and contributes to the field. The study well done with many supporting analyses and well written.

Our ref: NATECOLEVOL-211215425A

11th February 2022

Dear Dr. Sogin,

Thank you for your patience as we've prepared the guidelines for final submission of your Nature Ecology & Evolution manuscript, "Sugars dominate the seagrass rhizosphere" (NATECOLEVOL-211215425A). Please carefully follow the step-by-step instructions provided in the attached file, and add a response in each row of the table to indicate the changes that you have made. Please also check and comment on any additional marked-up edits we have proposed within the text. Ensuring that each point is addressed will help to ensure that your revised manuscript can be swiftly handed over to our production team.

3****We would like to start working on your revised paper, with all of the requested files and forms, as soon as possible (preferably within two weeks). Please get in contact with us immediately if you anticipate it taking more than two weeks to submit these revised files.****

In recognition of the time and expertise our reviewers provide to Nature Ecology & Evolution's editorial process, we would like to formally acknowledge their contribution to the external peer review of your manuscript entitled "Sugars dominate the seagrass rhizosphere". For those reviewers who give their assent, we will be publishing their names alongside the published article.

Nature Ecology & Evolution offers a Transparent Peer Review option for new original research manuscripts submitted after December 1st, 2019. As part of this initiative, we encourage our authors to support increased transparency into the peer review process by agreeing to have the reviewer comments, author rebuttal letters, and editorial decision letters published as a Supplementary item. When you submit your final files please clearly state in your cover letter whether or not you would like to participate in this initiative. Please note that failure to state your preference will result in delays in accepting your manuscript for publication.

Cover suggestions

As you prepare your final files we encourage you to consider whether you have any images or illustrations that may be appropriate for use on the cover of Nature Ecology & Evolution.

Nature Ecology & Evolution has now transitioned to a unified Rights Collection system which will allow

4our Author Services team to quickly and easily collect the rights and permissions required to publish your work. Approximately 10 days after your paper is formally accepted, you will receive an email in providing you with a link to complete the grant of rights. If your paper is eligible for Open Access, our Author Services team will also be in touch regarding any additional information that may be required to arrange payment for your article.

Please note that *Nature Ecology & Evolution* is a Transformative Journal (TJ). Authors may publish their research with us through the traditional subscription access route or make their paper immediately open access through payment of an article-processing charge (APC). Authors will not be required to make a final decision about access to their article until it has been accepted. [Find out more about Transformative Journals](https://www.springernature.com/gp/open-research/transformative-journals)

Authors may need to take specific actions to achieve compliance with funder and institutional open access mandates. For submissions from January 2021, if your research is supported by a funder that requires immediate open access (e.g. according to [Plan S principles](https://www.springernature.com/gp/open-research/plan-s-compliance)) then you should select the gold OA route, and we will direct you to the compliant route where possible. For authors selecting the subscription publication route our standard licensing terms will need to be accepted, including our [self-archiving policies](https://www.springernature.com/gp/open-research/policies/journal-policies). Those standard licensing terms will supersede any other terms that the author or any third party may assert apply to any version of the manuscript.

[REDACTED]

[REDACTED]

Author Rebuttal to Initial comments

5Reviewer #2 (Remarks to the Author):

This manuscript asserts that phenolics inhibit microbial sucrose metabolism within anoxic sediments under seagrass meadows leading to accumulation. This is an interesting and important topic. The manuscript is well-written with a compelling narrative and attractive figures. I have reviewed this manuscript previously and am very pleased to see that the authors have prepared a substantially improved manuscript in response to earlier reviews. A few issues remain as described below.

We thank reviewer #2 for all the time and effort they put into reviewing our manuscript. Please see our reply below where we address the few remaining issues.

1. Figure S2 is nice and the same should be included for the other sugars in S1.

We thank the reviewer for their complement and have now included the same validation figures for other sugars in Figure S1.

2. Figure S7 I'm not clear how you are differentiating caffeic vs. chicoric acids given that they appear to have the same retention time on your GC-MS method, you apparently see the same parent ion m/z 396 for both and in the caption you state they have similar fragmentation spectra. I would expect chicoric acid to elute after caffeic acid. Might it be the peak at 25min? What am I missing?

*We agree with the reviewer that we are unable to differentiate caffeic acid from chicoric acid using GC-MS, as shown in Figure S7 with pure standards. Chicoric acid is a compound made up of two molecules of caffeic acid connected to one molecule of tartaric acid via ester bonds that are not stable. Because these bonds are unstable, during sample derivatization chicoric acid degrades into caffeic acid¹. Therefore, the GC-MS cannot differentiate between the two phenolic compounds. Given that past studies using LC-MS have shown that *P. oceanica* tissues contain both chicoric acid and caffeic acid², we considered the caffeic acid peak from our *P. oceanica* extracts to represent the sum of caffeic acid and chicoric acid. We clarified this in the main text by modifying line 175 to read:*

6“P. oceanica contains phenolic compounds in its root tissues, including chicoric acid and caffeic acid (Extended Data 4; Figure S6).”

We have also include a description to explain in the legend for Figure S6 clarifying our findings. The legend for Figure S6 now includes the statement:

“Chicoric acid is a compound made up of two molecules of caffeic acid connected to one molecule of tartaric acid via ester bonds that are not stable. Because these bonds are unstable, during sample derivatization chicoric acid degrades into caffeic acid¹. Therefore, the GC-MS cannot differentiate between the two phenolic compounds. Given that past studies using LC-MS have shown that P. oceanica tissues contain both chicoric acid and caffeic acid², we considered the caffeic acid peak from our P. oceanica extracts to represent the sum of caffeic acid and chicoric acid.”

1. Olivier, D.; Costa, J.; Desjobert, J.-M.; Pergent, G., Variations in the concentration of phenolic compounds in the seagrass *Posidonia oceanica* under conditions of competition. *Phytochemistry* **2004**, *65* (24), 3211-3220.
2. Grignon-Dubois, M.; Rezzonico, B., Phenolic fingerprint of the seagrass *Posidonia oceanica* from four locations in the Mediterranean Sea: first evidence for the large predominance of chicoric acid. *Botanica Marina* **2015**, *58* (5), 379-391.
3. Figure S3b caption is unclear “A subset of the DOC samples collected inside the meadow show that sugars made up to 40% of the DOC composition within the 5 cm depth, where seagrass roots dominated the sediment.” This figure is very important to support the importance of sucrose to the overall global carbon stock (this could be in the main text). However, it is unclear what ‘A subset of DOC samples’ means? How were the samples selected?

Thank you for pointing this out. For each sampling depth, samples (n=3) were randomly selected for both DOC and GC-MS analysis. It was beyond the scope of our study to run subsets of all DOC samples using the GC-MS method. We have adjusted the text in the legend

of Figure S3b to reflect that the samples were randomly selected to calculate the percent contribution of sugars to the DOC.

4. I also couldn't find the DOC data. Figure S7 refers to table S2 but the tab/table labeled 'table s2' provides the 13C incubation data.

We thank the reviewer for catching the missing DOC data from the Supplement, we have now included it as Table S4. We have now adjusted all supported data tables and renamed them throughout the text.

5. Double check all of the figure and table references. For example line 177 DOM of *P. oceanica* pore waters was highest underneath the seagrass meadow between 5 and 25 cm sediment depth (Extended Data 4b; see Fig. S8 for identification caffeic acid in pore waters). Yet ED 4b shows the phenolics data. ED 3 fig3b shows DOC values.

We thank the reviewer for pointing out that some of the figures and tables need to be cross checked. We have now gone through the manuscript and supporting data to correct any missing references. With regards to line 177 and Extended Data Figure 4b (now Figure 3), this does in fact show the ion count for the molecular formula, $C_9H_8O_4$, which we extracted from the dissolved organic matter profiles (not the dissolved organic carbon concentrations). As originally written, we recognize the confusion and have re-written line 177 to read:

*"Indeed, the sum formula for caffeic acid ($C_9H_8O_4$) had highest counts in the pore waters underneath the seagrass meadow between 5 and 25 cm sediment depth (**Figure 3b**; **Table S2**; see **Figure S7** for identification of caffeic acid in sediment pore waters)"*

6. Line 176 to 179: Did you perform statistical analysis to support the assertion that caffeic acid abundance between different depths? This is required to support this assertion.

We thank the reviewer for this comment as we had not preformed the statistical test comparing ion abundances of this molecular formulae ($C_9H_8O_4$). We have now done so and have included the resulting statistics from the ANOVA model and post-hoc tests in the

appropriate supporting tables (for ANOVA models see Table S2 and for post-hoc tests see Table S3). These results support our findings that the count of caffeic acid is significantly higher inside the meadow than at the edge or outside the meadow. Furthermore, the two-way ANOVA model shows that the abundance is significantly higher between 5 and 25 cm below the sediment surface.

7. Lines 312-316 and related text. There isn't direct evidence that these clades of microbes are using sucrose so statements like "The three sucrose specialists that used sucrose inside the meadow" are inferences. Suggest rewording to something along the lines of "based on the genomic analysis we predict that it is these three sucrose specialists that used sucrose...."

We agree with the reviewer and have adjusted the text as follows:

Line 312 now reads: "Based on our genomic analyses, we predict that these three bacterial species preferentially metabolize sucrose over other sugars. These three putative sucrose specialists were undescribed members...."

Line 316 now reads: "The putative sucrose specialist at the edge of the meadow.."

8. The 'n' number (number of replicates) is missing from most of the figures (e.g. Fig 1b,2a,2b,4b,S1,S3)

We thank the reviewer for pointing this out. We have included replication numbers in Figure 1b in the plot itself. For all other figures, we now report the replication numbers in the figure legend (Fig 2a, 2b, 4b, S3) and in Table S3 (Figure S1).

9. You should perform hoc test for ANOVA is missing for example using hoc Tukey test for results showed in Fig 1b, 4b, S1, S3a, S3b,S4, S6.

We agree with the reviewer that it is important to perform the post-hoc test for all ANOVA models. Post-hoc tests were run for all data presented in the figures requested by the reviewer

except for Figure S3b (percent of DOC data) as these data were not suitable for statistical analysis and it was beyond the scope of our study to compare the percent contribution of sugars to the DOC across sampling depths. Because of the complexity of contrasts in the two-way ANOVA models, we have chosen to report the results of the post-hoc tests in Table S3.

Minor comments:

10. Currently it is very hard to find the SI tables. All supplementary tables should have the name of the table on the corresponding tab of the excel file. The use of both SI tables and figures as well as extended data makes this even more confusing. Is there a need to have both extended data and SI? For example “DOM of *P. oceanica* pore waters was highest underneath the seagrass meadow between 5 and 25 cm sediment depth (Extended Data 4b; see Fig. S8 for identification caffeic acid in pore waters).

We thank the reviewer for pointing this out. We have now re-organized all supporting tables and figures into a single excel sheet with individual tabs for each table. We agree that this will help the reader in finding the appropriate dataset. As far as the extended data is concerned, we choose to keep the format as it is as the extended data help highlight the important aspects of the manuscript and the SI provides essential supporting information to our findings. Finally, we have performed detailed checks to ensure all data links through the manuscript and supplement are correct.

Reviewer #3 (Remarks to the Author):

The authors have done a careful revision of the manuscript based on the comments from the reviewers. There were many questions/comments and I find that the manuscript now appears in a much more reflected version taken into account the uncertainties of the study. The results of the study are exciting and contributes to the field. The study well done with many supporting analyses and well written.

We thank the reviewer for the time and effort they put into reviewing our paper and helping us improve the quality of our manuscript. Thank you for your positive comments.

Final Decision Letter:

21st March 2022

Dear Dr Sogin,

We are pleased to inform you that your Article entitled "Sugars dominate the seagrass rhizosphere", has now been accepted for publication in Nature Ecology & Evolution.

Over the next few weeks, your paper will be copyedited to ensure that it conforms to Nature Ecology and Evolution style. Once your paper is typeset, you will receive an email with a link to choose the appropriate publishing options for your paper and our Author Services team will be in touch regarding any additional information that may be required

You will not receive your proofs until the publishing agreement has been received through our system

Due to the importance of these deadlines, we ask you please us know now whether you will be difficult to contact over the next month. If this is the case, we ask you provide us with the contact information (email, phone and fax) of someone who will be able to check the proofs on your behalf, and who will be available to address any last-minute problems . Once your paper has been scheduled for online publication, the Nature press office will be in touch to confirm the details.

Acceptance of your manuscript is conditional on all authors' agreement with our publication policies (see www.nature.com/authors/policies/index.html). In particular your manuscript must not be published elsewhere and there must be no announcement of the work to any media outlet until the publication date (the day on which it is uploaded onto our web site).

Please note that *Nature Ecology & Evolution* is a Transformative Journal (TJ). Authors may publish their research with us through the traditional subscription access route or make their paper immediately open access through payment of an article-processing charge (APC). Authors will not be required to make a final decision about access to their article until it has been accepted. [Find out more about Transformative Journals](https://www.springernature.com/gp/open-research/transformative-journals)

Authors may need to take specific actions to achieve [compliance with funder and institutional open access mandates](https://www.springernature.com/gp/open-research/funding/policy-compliance-faqs). If your research is supported by a funder that requires immediate open access (e.g. according to [Plan S principles](https://www.springernature.com/gp/open-research/plan-s-compliance)) then you should select the gold OA route, and we will direct you to the compliant route where

11possible. For authors selecting the subscription publication route, the journal's standard licensing terms will need to be accepted, including <https://www.nature.com/nature-portfolio/editorial-policies/self-archiving-and-license-to-publish>. Those licensing terms will supersede any other terms that the author or any third party may assert apply to any version of the manuscript.

We welcome the submission of potential cover material (including a short caption of around 40 words) related to your manuscript; suggestions should be sent to Nature Ecology & Evolution as electronic files (the image should be 300 dpi at 210 x 297 mm in either TIFF or JPEG format). Please note that such pictures should be selected more for their aesthetic appeal than for their scientific content, and that colour images work better than black and white or grayscale images. Please do not try to design a cover with the Nature Ecology & Evolution logo etc., and please do not submit composites of images related to your work. I am sure you will understand that we cannot make any promise as to whether any of your suggestions might be selected for the cover of the journal.

You can generate the link yourself when you receive your article DOI by entering it here: <http://authors.springernature.com/share>.

[REDACTED]

P.S. Click on the following link if you would like to recommend Nature Ecology & Evolution to your

12librarian <http://www.nature.com/subscriptions/recommend.html#forms>

** Visit the Springer Nature Editorial and Publishing website at http://editorial-jobs.springernature.com?utm_source=ejp_NEcoE_email&utm_medium=ejp_NEcoE_email&utm_campaign=ejp_NEcoE for more information about our career opportunities. If you have any questions please click [here](mailto:editorial.publishing.jobs@springernature.com).**